# Recent Developments in Seafood Packaging Technologies

**DOI:** 10.3390/foods10050940

**Published:** 2021-04-25

**Authors:** Michael G. Kontominas, Anastasia V. Badeka, Ioanna S. Kosma, Cosmas I. Nathanailides

**Affiliations:** 1Department of Chemistry, Faculty of Natural Sciences, University of Ioannina, 45110 Ioannina, Greece; abadeka@uoi.gr (A.V.B.); i.kosma@uoi.gr (I.S.K.); 2Faculty of Agriculture, University of Ioannina, 47100 Arta, Greece; nathan@uoi.gr

**Keywords:** seafood spoilage, packaging technologies, packaging legislation

## Abstract

Seafood products are highly perishable, owing to their high water activity, close to neutral pH, and high content of unsaturated lipids and non-protein nitrogenous compounds. Thus, such products require immediate processing and/or packaging to retain their safety and quality. At the same time, consumers prefer fresh, minimally processed seafood products that maintain their initial quality properties. The present article aims to review the literature over the past decade on: (i) innovative, individual packaging technologies applied to extend the shelf life of fish and fishery products, (ii) the most common combinations of the above technologies applied as multiple hurdles to maximize the shelf life of seafood products, and (iii) the respective food packaging legislation. Packaging technologies covered include: Modified atmosphere packaging; vacuum packaging; vacuum skin packaging; active food packaging, including oxygen scavengers; carbon dioxide emitters; moisture regulators; antioxidant and antimicrobial packaging; intelligent packaging, including freshness indicators; time–temperature indicators and leakage indicators; retort pouch processing and edible films; coatings/biodegradable packaging, used individually or in combination for maximum preservation potential.

## 1. Introduction

Among food commodities, seafood products (fish, crustaceans, mollusks) rank high in commercial value due to their high nutritional value as well as to their specific sensory characteristics upon cooking. The European fishery product industry amounted to ca. EUR 18 billion in 2003 [1]. Fish and fishery products contain protein of high biological value, vitamins, minerals and unsaturated fats which are beneficial to health. However, mainly due to their high water activity, close to neutral pH, and specific composition, seafood products are highly perishable; thus, they require immediate processing and/or packaging to retain their safety and quality. At the same time, consumers demand fresh seafood products that have undergone minimal processing, maintaining their initial high quality characteristics. In this respect, packaging comprises a critical factor for the seafood industry in order to ensure that products reach short or long distance destinations in the highest quality, while ensuring safety for consumers. The present review article focuses on innovative packaging technologies and their application for seafood, including basic information on the spoilage of seafood products as well as the respective legislation that covers packaging requirements.

## 2. Composition and Structure of Seafood 

The term ‘seafood’ covers: (1) pelagic and fresh water fish; (2) mollusks (clams, oysters, scallops, squid); (3) crustaceans (crab, lobster, shrimp, crayfish); (4) respective aquacultured species. Fish may be further categorized as: (i) fish from cold or warm water; (ii) salt water vs. fresh water fish; (iii) low fat vs. high fat fish; (iv) free swimming vs. aquacultured fish. Water temperature significantly affects the spoilage patterns of seafood, as warm water fishery products harbor a higher number of bacteria than their cold water counterparts. Furthermore, cold water fish harbor predominantly psychrotrophic bacteria, while in warm water fish, mesophilic bacteria predominate. In regard to differences in bacterial populations between salt and fresh water fish, it is evident that bacteria associated with marine fish should be tolerant to sea water salt concentrations (2–3%). 

Despite similarities in the structure and composition of mammalian meat and fish, fish exhibit distinctive differences compared to meat. Firstly, fish have no visually perceptible fat deposits. The fat content in fish ranges from very low (<1%) to over 20% *w*/*w* but is to a large extent dispersed between the muscle fibers, in contrast to mammalian meat. A second difference is that the collagen in fish is ca. 3%, whereas in mammalian meat it may reach 15%. A third difference is the structure of fish muscle. The muscle in mammalian meat consists of long muscle fibers, whereas the respective muscle fibers in fish are made up of short segments known as ‘myotomes’. In turn, myotomes are enveloped within collagen macromolecules known as ‘myocommata’ [2]. This specific structure of fish muscle results in the characteristic flaky texture of fish flesh. A final difference of fish compared to meat muscle is that not all nitrogenous compounds in fish are in the form of proteins. Non-protein nitrogen containing compounds (NPNC) consist of free amino acids, ammonia, di- and trimethylamine (known as volatile nitrogen bases, TVB-N), trimethylamine oxide (TMAO), carnosine, creatine, histamine, etc. Examples of NPNC content includes cod, with a protein nitrogen content of 2.47%/total N = 2.83% and lobster, with a protein nitrogen content of 2.04%/total N = 2.72%. Shewan [3] reported that approximately 1.5% of fish muscle consists of non-protein soluble components.

Depending on the species, size and fishing grounds, the composition of fish muscle varies significantly. Specific diet also affects fish muscle composition in aquacultured fish [4]. For example, cod (non-fatty fish) is composed of 81.5% moisture, 16.5% protein, fat 0.4%, 0% carbohydrate and 1.2% ash, whereas salmon (fatty fish) is composed of 63.5% moisture, 17.5% protein, 16.5% fat, 0% carbohydrate and 1% ash. 

## 3. Seafood Spoilage 

The muscle of freshly captured fish is sterile, as is the case for mammals and birds. In contrast, the skin, gills and intestines have a medium to very high bacterial load, 10^2^–10^7^ cfu/cm^2^ for the skin and 10^3^–10^9^ cfu/g for the gills and intestines. The fish microbiota from northern temperate waters consist mainly of psychrotrophic and psychrophilic Gram(-) pseudomonads and H_2_S producing bacteria, including *Shewanella putrefaciens*. *Psychrobacter, Vibrio, Acinobacter and Moraxella* spp. may also be part of the microbiota. Fish from warm, tropical waters usually harbor fewer psychrotrophic bacteria with spoilage potential, resulting in longer keeping times of these fish compared to those from colder waters [2]. In addition to the above, it has been reported that *Photobacterium phosphoreum* is the dominant spoilage microorganism in seafood products such as haddock fillets kept refrigerated or at higher temperatures. *Pseudomonas* spp. and *S. putrefaciens* also comprise part of the spoilage microbiota responsible for the development of off-odors [5]. 

During the beginning of spoilage, bacteria use non-protein nitrogen to form ammonia, dimethyl and trimethylamine. The following stage involves proteolysis, during which, spoilage becomes more evident. Sulfur compounds, including H_2_S, mercaptans and (CH_3_)_2_S are the products of the action of *S. putrefacians* and also some pseudomonads, contributing to spoilage [4]. Upon harvesting at sea or in the fish farm, fish are either fresh-frozen or stored on ice. Subsequent processing, such as filleting and mincing, also increases the spoilage rate of fish as a result of increasing the surface/volume of the product. Fish can be further contaminated during the various stages of processing by pathogens including *Vibrio cholerae*, *Vibrio parahaemolyticus Clostridium botulinum Type E*, enteric viruses and toxins, causing scombroid fish poisoning and paralytic shellfish poisoning. 

As regards mollusks, their flesh differs from that of finfish and crustaceans as they contain an appreciable amount of glycogen. Thus, the spoilage of mollusks is primarily glycolytic (1–5% glycogen content), rather than proteolytic as is the case with finfish and crustaceans. The decomposition of glycogen results in a pH ca. 6.5–5.8, which is usually lower than that of finfish and crustaceans. Under such conditions, the microflora of mollusks are the same during the early stages of spoilage as those of finfish and crustaceans but they differ during the later stages of spoilage, where enterococci, lactobacilli and yeasts dominate due to prevailing lower pH values. For this reason, mollusks are usually transported live to the point of sale, where they then undergo further processing.

In turn, crustaceans including shrimp and prawns, apart from their higher pH, are often contaminated with bacteria from the mud, accompanying them during harvesting, resulting in more rapid microbiological deterioration upon capture [1]. As regards their microbial spoilage patterns, crustaceans are very similar to fish; the difference being the higher content of free amino acids and other soluble nitrogen containing compounds, leading to more rapid spoilage due to elevated levels of TVB-N compounds being produced [4]. In general, seafood spoilage is the combined result of (i) enzymic autolysis, (ii) microbial proliferation and (iii) lipid oxidation. 

### 3.1. Spoilage Due to ENZYMIC Autolysis 

The dominant changes that occur in fish flesh immediately after slaughter are the result of enzyme activity [6]. Autolytic enzymes in the flesh of seafood products are highly active, leading to the autolysis of fish muscle, which occurs more rapidly in seafood compared to mammals and poultry. The reason for this is that seafood pH is usually between 6.2 and 6.5 compared to 5.5–5.8 in mammalian muscle. Autolysis does not affect product odor and flavor but considerably affects the texture of fish muscle [7,8]. Enzymes such as proteases and lipases decompose proteins and fats, respectively, during fish spoilage [9].

Among the various quality indices used to evaluate the degree of freshness of fish, the K-value is a biochemical index based on adenine nucleotide catabolism reactions. This set of reactions involves: ATP → ADP → AMP → IMP → inosine → hypoxanthine → uric acid, solely through enzymic action. K is defined as the ratio: (inosine) + (hypoxanthine)**/**(ATP) + (ADP) + (AMP) + (IMP) + (inosine) + (hypoxanthine) × 100. In the early stages of spoilage, the Κ-value remains low at a value around 10%, increasing with time and endogenous enzyme activity. As microbial activity dominates the spoilage mechanism, K exponentially increases. K-values below 20% are indicative of optimal fish quality. K-values between 20 and 40% indicate acceptable fish quality, while a K-value higher than 40% corresponds to unacceptable fish quality. It has been documented that changes in the K-value show a good correlation with the degree of spoilage [10]. Ahmad et al. [11] determined the K-value in a study involving the evaluation of sea bass fillet quality. Fish were wrapped in films made of gelatin containing 25% lemongrass essential oil and refrigerated. The microbiological, chemical and physical parameters were monitored over time. The controls were considered microbiologically unacceptable on day 10 of storage (K-value: 66%), while test samples recorded lower K-values until day 12 of storage.

### 3.2. Microbial Spoilage 

As stated above, and depending on the type of aqueous environment fish live in (cold vs. warm waters, salt vs. fresh water, etc.), the microbial load of fish is made up of Pseudomonas, H_2_S producing bacteria, *Moraxella, Alcaligenes, Vibrio, Serratia* and *Micrococcus* spp. [12]. Microbial proliferation and enzymic activity are mainly responsible for fish spoilage, leading to the formation of a large number of volatile compounds (alcohols, carbonyl compounds, organic acids, sulfur and nitrogen compounds, etc.) responsible for the development of characteristic unpleasant and unacceptable off-odors and flavors [13]. While raw unpreserved fish usually spoil through the action of Gram(−), fermentative bacteria (i.e., Vibrionaceae), chilled fish spoil via psychrotrophic Gram(−) bacteria, i.e., pseudomonads and *Shewanella* spp. Thus, it is of key importance to distinguish between microbiota included in the total viable count (TVC) and specific spoilage organisms (SSO), which actually cause fish spoilage [14]. As a result of bacterial growth, protein breakdown occurs, leading to the formation of trimethylamine (TMA). Thus, the concentration of TMA formed is a good indicator of the degree of fish spoilage. Furthermore, trimethylamine oxide (TMAO) occurs in considerable quantities in marine fish as part of their osmoregulatory system. TMAO is used by numerous spoilage bacteria such as *S. putrefaciens*, psychrotolerant Enterobacteriaceae, *Aeromonas* spp. *P. phosphoreum* and *Vibrio* spp. as an energy source [15]. Upon reduction through the action of TMO reductase, TMA is formed. The formation of TMA is accompanied by the production of ammonia-like off-flavors. It is noteworthy to mention that the TMAO content varies significantly depending on the marine species. 

Another widely used chemical index to determine seafood quality is TVB-N. TVB-N includes ammonia, dimethylamine and trimethylamine, all being the products of amino acid decarboxylation through the action of decarboxylase enzymes, produced during protein decomposition. Connell [15] proposed TVB-N values between 30–35 mg TVB-N/100 g as the upper limit of acceptability for specific seafood family products. Recently, besides TVB-N and TMA (used to monitor microbial spoilage of fish), volatile organic compounds (VOCs) have also been used as indicators of fish quality [16]. Methods used to determine VOCs include static headspace, dynamic headspace and solid phase micro extraction coupled to gas chromatography/mass spectrometry (SPME-GC/MS) techniques. Among these, SPME-GC/MS has been very successfully used to assess seafood quality, correlating well with microbiological data [17]. 

In the intermediate stages of spoilage, SSO such as *Pseudomonas fluorescens*, *S. putrefaciens*, and others grow rapidly. The products of their metabolism include proteases and lipases which, in turn, decompose proteins and fats, respectively. As spoilage further proceeds, proliferation of SSO leads to the sensory rejection of fish due to the formation of objectionable odors and flavors [18]. 

Another quality criterion for fish and fishery products is the content of biogenic amines and more specifically histamine. Biogenic amines (BAs) are low molecular weight nitrogenous organic compounds that are polar or semi-polar in nature. The chemical structures of BAs are aliphatic (putrescine, cadaverine, spermine, spermidine), aromatic (tyramine, 2-phenylethylamine), or heterocyclic (histamine, tryptamine). BAs are usually formed by the decarboxylation of amino acids or by amination and trans-amination of aldehydes and ketones [19]. Therefore, seafood rich in protein, exposed to bacterial action and/or fermentation and/or following time/temperature abuse could induce higher contents of BAs. Gram(+) and Gram(−) bacteria related with fish spoilage can produce BAs. The most common species are mesophilic and psychrotolerant bacteria of the Enterobacteriaceae family such as *Morganella, Enterobacter, Hafnia, Proteus*, and *Photobacterium*. Likewise *Pseudomonas* spp. and lactic acid bacteria (LAB), belonging to the *Lactobacillus* and *Enterococcus* genera, can form BAs [20]. High concentrations of histamine are the cause of histamine fish poisoning, or Scombroid poisoning, resulting in symptoms similar to those related to seafood allergies. Symptoms include a metallic taste sensation, oral numbness, headache, dizziness, low blood pressure, difficulty in swallowing, itchy skin, and rash [21]. 

According to the FDA guidelines and EU Regulations the toxicity and defect action levels for histamine in *Scombridae*, *Scombresosidae*, *Clupeidae*, *Engraulidae*, *Coryfenidae*, *Pomatomidae* fish families have been established to be 50 and 100 mg/kg, respectively [22]. Finally, it should be emphasized that fish spoilage does not usually concur with the presence of pathogenic microorganisms or their metabolic products (toxins). The first relates to fish quality, while the second refers to safety issues.

### 3.3. Oxidation and Hydrolysis

Lipid oxidation is a common deteriorative chemical process which occurs mostly in fish species of high fat content, i.e., salmon, mackerel, sardines, etc. It is the reaction of double bonds in fish triglycerides with oxygen. Fish lipids contain up to 40% of aliphatic long-chain fatty acids (C14–C22) which are polyunsaturated. While mammalian meat fat mostly contains fatty acids with two double bonds, fish fat contains a number of fatty acids with five or six double bonds. Moreover, fish oils contain other polyunsaturated fatty acids (PUFA), i.e., eicosapentaenoic (EPA, C20:5n3) and docosa-hexaenoic (DHA, C22:6n3) acids which are considered as ‘essential fatty acids’ [23]. Initial oxidation products include hydroperoxides, which, as unstable compounds, eventually break down to form mainly aldehydes and ketones which comprise the secondary products of oxidation. Malondialdehyde (MDA) is the aldehyde measured by the thiobarbituric acid (TBA) test and is directly related to the degree of fish lipid oxidation. Secondary oxidation products are the source of characteristic rancid off-odors and flavors of oxidized lipids. In order for oxidation to proceed, molecular oxygen must be activated to its singlet oxygen state. Activation of oxygen molecules is usually achieved in the presence of transition metals [24]. Compounds serving as catalysts for the initial reaction of fatty acid double bonds with singlet state oxygen include myoglobin, hemoglobin and cytochrome [9]. 

On the other hand, hydrolysis in fish fat occurs through enzymic and non-enzymic reactions. Lipolysis refers to the enzymic hydrolysis of fats by lipases. In this reaction, lipase enzymes decompose triglycerides to free fatty acids and glycerol. The former increase the titratable acidity of fish oil, and thus, decrease the quality of fish oil. The enzymes involved in lipolysis (lipases) can either be the products of the metabolism of psychrotrophic spoilage microorganisms or endogenous of the fish itself [25]. Lipases are present in fish skin, blood and tissues. The reaction of fatty acids liberated during the hydrolysis of fish lipids with sarcoplasmic and myofibrillar fish proteins results in protein denaturation. According to Undeland et al. [26], the highly pro-oxidative hemoglobin pigment can cause lipid oxidation in fish muscle. Oxymyolobin (Fe^2+^) and metmyoglobin (Fe^3+^) have a higher pro-oxidative potential. 

Besides spoilage microorganisms, seafood also harbor numerous food pathogenic microorganisms. *Salmonella* spp., being the major cause of bacterial illness in seafood products, grows within a wide range of temperatures between 5.2–47 °C and pH values between 3.7 and 9.5. Despite this, several *Salmonella* strains have been documented to survive at temperatures below 0 °C for up to 9 months [27]. *Listeria monocytogenes* is another food pathogen that commonly occurs in seafood processing facilities, although it is less widespread than *Salmonella* spp. *L. monocytogenes* also grows at temperatures in the range of 0.4–45 °C and achieves high counts in seafood products such catfish and shrimp. Vibrio spp. is yet another common seafood pathogen. *Vibrio vulnificus*, the most virulent Vibrio species, has been repeatedly associated with gastroenteritis. In a study conducted by Schwarz [28] oysters which underwent immediate cooling exhibited a 97.8% decrease in the population of *V. vulnificus*, compared to commercially cooled oysters which reached the same result after four days of storage. The main microbiological concern of the seafood industry is *C. botulinum*, a spore-forming strictly anaerobic bacterium that grows just above 3.3 °C. Even today, the heat stability and high toxicity of its toxin constitute a challenge for seafood processors. Another pathogen commonly associated with seafood is *Aeromonas hydrophila*, found in particular in finfish and prawns. In a survey study on Greek seafood by Papadopoulou et al. [29], *A. hydrophila* was found to be the predominant contaminant, comprising 38% of contaminant organisms in freshwater fish, 73–86% in shellfish and 93% in pelagic finfish 24 h after harvesting.

Finally, among enteric viruses responsible for human illness incidents, Norovirus (NoV) has been associated with the consumption of bivalve mollusks. NoV is the cause of mild gastroenteritis, often accompanied with nausea, diarrhea, abdominal pain, vomiting, and fever. The incubation period is 1 to 4 days. Symptoms last for ca. 2 days, after which, complete patient recovery occurs.

## 4. Seafood Packaging Technologies 

According to the former Packaging Institute International [30], packaging is defined as the enclosure of products in a pouch, bag, box, cup, tray, can, tube, bottle or other form of container serving the following functions: containment, protection, preservation, utility and communication. The role of packaging becomes more critical when it deals with foods and pharmaceutical products that, through the GI tract, come in contact with the internal organs of the body, thus, directly affecting human health. Among the above mentioned functions of packaging, protection and preservation are the most important. With regard to the protection objective, the package protects the food contents from (i) physico-chemical and (ii) microbiological changes which result in reductions in product quality and safety and (iii) physical/mechanical damage [31]. The preservation function of packaging is the result of the protection it provides to the contained product from the effect of oxygen, light, moisture, odors and biological contamination by pathogenic microorganisms and pests. Along the same line of reasoning, the function of packaging becomes even more critical when it comes to seafood products, which are extremely perishable commodities prone to both spoilage and contamination by pathogenic microorganisms [32]. More recently, along with advances in technology, packaging has transformed from a simple passive barrier to a dynamic interactive system between the contained food and the packaging material, reflected in technologies known as modified atmosphere packaging, vacuum packaging, active packaging, intelligent packaging, etc. All these technologies will be further discussed as applied to fish and fishery products.

### 4.1. Modified Atmosphere Packaging 

Modified atmosphere packaging (MAP) may be defined as the packaging of food in an atmosphere, the composition of which is initially modified using specific fixed gases, according to a desired profile. MAP inhibits the oxidation of lipids, aerobic bacterial growth and extends the shelf life of the product. The product is usually packaged in a barrier material along with a mixture of gases, the composition of which depends on the product properties, the expected shelf life, the packaging material permeability properties and the storage conditions. Temperature is a very important parameter in MAP. This technology usually works well at reduced temperatures. During storage, the mixture of gasses within the package can change due to the respiration rate of the packaged food, the specific permeability of the packaging material to fixed gases, time and temperature. O_2_, CO_2_ and N_2_ are most often used in MAP commercial applications even though CO, SO_2_, C_2_H_4_, etc., have been also used mostly in research studies. The former are used in different combinations and concentrations depending on the nature and properties of the product to be packaged. The choice depends on the type of microbial flora capable of growing on the product, the sensitivity of the product to oxygen and carbon dioxide, and color stability requirements, i.e., for low fat, white fish (sea bream, sea bass, catfish, cod, croaker, flounder, haddock, hake, halibut, mullet, red snapper, turbot, whiting) a gas mixture of 30% O_2_/40% CO_2_/30% N_2_ is recommended for retail packaged fish. The same holds for shellfish, including crustaceans and mollusks (clams, cockles, crab, crayfish, cuttlefish, lobster, mussels, octopus, oysters, prawns, scallops, sea urchins, shrimp, squid). The respective gas mixture for fatty fish is 40% CO_2_/60% N_2_. Of the gases used, nitrogen is an inert gas used to inhibit oxidation in oxygen-sensitive foods; carbon dioxide functions as an antimicrobial agent, forming carbonic acid (H_2_CO_3_) in aqueous media which, in turn, reduces the pH of the product, resulting in product shelf life extension [33]. This provides a substantial advantage, particularly to perishable food products, by prolonging the distribution chain and diminishing the need for centralized production. Between the two major groups of foods to which MAP is successfully applied (muscle foods and fresh produce), the former comprises an excellent substrate for the growth of both spoilage microorganisms and pathogens, directly affecting both food product quality and safety. As regards oxygen, MAP usually maintains an atmosphere poor in oxygen. Such an atmosphere greatly inhibits the growth of strict aerobes, i.e., the pseudomonads, which comprise the major spoilers of muscle foods (meat, fish, poultry). This, however, poses a potential danger where the development of anaerobic pathogens is possible. *C. botulinum* is a strict anaerobe, while, additionally, non-proteolytic *C. botulinum* type E is psychrotropic, capable of producing its deadly toxin under refrigerated conditions (at temperatures above 3.3 °C). In such cases, the concentration of O_2_ maintained within the package in combination with the oxygen permeability of the packaging material is the result of a compromise between the inhibition of oxidation and the potential risk of extreme anaerobiosis [32]. 

The MAP of meat was first reported in a shipment of Australian lamb carcasses to England in the 1930s and has enjoyed success worldwide during the past 40 years [34]. The shelf life and safety of MAP foods is mainly affected by: (i) the nature of the food, (ii) the mixture of gases within the package, (iii) the gas permeability of materials used and (iv) temperature. Given the general modern trend for the consumption of minimally processed, non-frozen chilled meats and fish, and ready-to-eat meals, the food industry has seen a fast-growing market for applying MAP. Modified atmosphere packaging is inexpensive, suitable and easily applied to several food products, packaging machinery and packaging materials. Extensive research on the subject has shown that MAP significantly prolongs the shelf life of food products, and in some cases, no other additional treatment is required during distribution. In most cases, however, the objectives of MAP are achieved through the use of the multiple hurdle principle. Hurdle technology is critical for most MAP applications, because the modified atmosphere provides an ‘unnatural’ gas environment which favors the growth of anaerobic bacteria and the production of microbial toxins. When combining an effective temperature control system along with MAP, product quality and safety are ensured. Another requirement for the effective application of MAP is the use of optimal packaging materials with specific gas permeability properties [33]. 

#### 4.1.1. Use of MAP for Fish Preservation

Speranza et al. [35] investigated the effect MAP (two different gas mixtures) on the quality retention of four ready-to-cook fishery products, namely, hake fillets, yellow gurnard fillets, chub mackerel fillets, and gutted cuttlefish. The first MAP used contained 40% CO_2,_ 30% O_2_ and 30% N_2_. The second contained 95% CO_2_ and 5% O_2_. Both MAP mixtures were effective in suppressing microbial proliferation and retaining organoleptic quality of the products. A shelf life extension of MAP samples between 95 and 250% was achieved compared to controls. Yesudhason et al. [36] combined MAP with sodium acetate for the shelf life extension of seer fish. The lowest TVC values were recorded for MAP samples pre-treated with sodium acetate. Control (air packaged) samples had a shelf life of 8 days, MAP samples 22 days and MAP (70% CO_2_/30% O_2_) plus sodium acetate (1%, *w*/*v*) had a shelf life of 28 days. Kykkidou et al. [37] investigated the effect of thyme essential oil and packaging on fresh Mediterranean swordfish fillets during storage at 4 °C. Treatments included: air (A), MAP (M), air with thyme oil (AT) and MAP with thyme oil (MT). Of all treatments, only MT inhibited product lipid oxidation during storage. Sensory evaluation showed a shelf life of 8 days for aerobically packaged refrigerated swordfish and 13 days for sword fish packaged under MA. Addition of 0.1% thyme essential oil increased the aerobically packaged product shelf life by 5 days, whereas combining MAP with thyme oil resulted in a 7–8 day product shelf life increase compared to the control sample. Del Nobile et al. [38] studied the combined effect of three essential oils (EOs)/extracts (thymol, lemon extract and grapefruit seed extract (GFSE)) and MAP on the quality of blue fish burgers. Samples were packaged in air and in three different gas mixture compositions: 30% O_2_/40% CO_2_/30% N_2_, 50% O_2_/50% CO_2_ and 5% O_2_/95% CO_2_. Burgers were stored at 4 °C for 28 days. The formation of biogenic amines was also evaluated. Results showed the retention of the microbiological quality of the product using a very small concentration of thymol (110 ppm), GFSE (100 ppm) and lemon extract (120 ppm) combined with MAP. Sensory analysis results showed a shelf life of 22–23 days for fish burgers treated with high CO_2_ MAP+ essential oil extracts, with no effect on product nutritional quality. Fernández et al. [39] studied the effect of natural additives, super-chilling and MAP on the shelf life of Atlantic salmon fillets. The parameters studied included: gas concentration (CO_2_/N_2_), gas-to-product volume (*g*/*p*) ratio and type of natural additive used. Natural additives had no effect on salmon shelf life. Combining super-chilling with MAP resulted in the greatest extension of shelf life. The highest CO_2_ concentration (90%) and *g*/*p* ratio of 2.5 resulted in the longest shelf life, that is, 22 days compared to 11 days for the controls. Torrieri et al. [40] investigated the combined effect of MAP and antioxidant packaging on the shelf life extension of fresh bluefin tuna fillets stored at 3 °C. Antioxidant films were prepared by incorporating α-tocopherol into an LDPE matrix at three different concentrations (0.1%, 0.5%, 1%). Preliminary shelf life tests were run to select the best gas composition and the best α-tocopherol concentration in the test film. Results showed that (i) 100% N_2_ atmosphere had a protective effect on color and lipid oxidation, (ii) antioxidant film reduced fat oxidation, (iii) MAP combined with the antioxidant film was able to extend the shelf life of raw fish products. The effect of MAP (MAP 1: 40% CO_2_/60% N_2_ and MAP 2: 100% CO_2_) on the quality and shelf life of carp steaks was studied by Babic et al. [41]. Carp steaks were stored at 3 °C up to 15 days. Sensory analysis (odor scores) showed that carp steaks packaged in MAP1 had a shelf life of 13 days, while those packaged in MAP2 remained acceptable until day 15 of storage. Messina et al. [42] investigated the effect of natural antioxidants (AOX) and MAP on the quality and shelf life of dolphin-fish fillets. Unpackaged fillets simply placed on trays served as controls. MAP fillets were preserved under an atmosphere of 45% CO_2_/50% N_2_/5% O_2_, and (MAP-AOX) fillets were preserved by combining natural antioxidants from halophytes with MAP. All samples were stored under refrigeration at −1 °C for 18 days. Sensory testing and color analyses showed that MAP and MAP-AOX fillets maintained the highest quality. Fatty acids (FA) were protected against oxidation only in the MAP and MAP-AOX groups. The sensory properties of the fillets correlated well to changes in FA content, peroxide value (PV) and MDA content. The bacterial growth and sensory quality of farmed Atlantic cod fillets were studied by Hansen et al. [43]. Samples were packaged either under vacuum (VP) conditions or in MAP (60% CO_2_/40% N_2_) with a CO_2_ emitter pad or a liquid absorbent pad. Fillets were stored for 15 days at 2°C. The study objective was to investigate the effect of packaging with a CO_2_ emitter for product shelf life extension under vacuum conditions, and in low headspace MAP. Sensory analysis showed that the initial quality of the samples was better preserved by adding a CO_2_ emitter under both VP and MAP. The shelf life of VP samples was 7 days, VP+ CO_2_ emitter and MAP, 9 days, and MAP + CO_2_ emitter, 13 days. Kuuliala et al. [44] identified and quantified volatile compounds (VC) indicative of spoilage in raw Atlantic cod packaged under MA (60% CO_2_/40% O_2_/0% N_2_), (60% CO_2_/5% O_2_/35% N_2_) and air, stored at 4 and 8 °C. 16S rRNA gene sequence analysis was used for the identification of the cod microbiota, while selected-ion flow-tube mass spectrometry (SIFT-MS) was used for the quantification of selected VC. The results showed that *Photobacterium* spp. was the main spoiler and VC producer. An increase in exponential VC concentration and sensory rejection occurred at high TVC (7–7.5 log), leading to the conclusion that monitoring of the early stages of spoilage calls for high method sensitivity for low VC concentrations. Tsironi et al. [45] developed and validated a kinetic model for the growth of spoilage bacteria in MAP gilthead sea bream fillets in order to design an effective time temperature integrator (TTI). The temperature and CO_2_ dependence of the growth of LAB in MAP fish fillets was expressed by an Arrhenius-type model in the range of 0–15 °C and 20–80% CO_2_. The model was validated under various temperature conditions. A UV activated, photochemical TTI, was developed and the effect of the level of activation on the response of the TTI was modeled. The TTI response was tailored to monitor the shelf life of fish fillets at selected MAP conditions, during chill chain storage. Parlapani et al. [46] investigated the spoilage microflora of sea bream fillets packaged aerobically and using MAP, stored at 0 and 5 °C, by 16S rRNA gene sequence analysis of isolates grown on plates. Sensory evaluation, determination of TVC and spoilage microflora was carried out to determine product shelf life and the bacterial growth profile. The results showed that the growth and synthesis of spoilage microbiota were affected by temperature and the specific atmosphere used. The same parameters affected shelf life. Product shelf life in air at 0 and 5 °C was 14 and 5 days, respectively. The respective shelf life under MAP was 20 and 8 days. At the end of shelf life, the dominant microflora was closely related to *Pseudomonas fragi*. *Pseudomonas veronii* dominated in fillets under MAP at 0 °C. Furthermore, *Carnobacterium maltaromaticum*, *Carnobacterium divergens* and *Vagococcus fluvialis* prevailed in fillets stored under MAP at 5 °C. Rodrigues et al. [47] studied the effect of UV-C radiation combined with MAP on the quality of refrigerated rainbow trout fillets during storage for 22 days. Sample treatments were: (i) aerobic packaging (AP), (ii) VP, (iii) VP + UV-C radiation (106.32 mJ/cm^2^), (iv) (MAP) (80% CO_2_/20% N_2_) and (v) (MAP + UV-C). The results showed that MAP inhibited microbial growth and retarded chemical changes, resulting in the shelf life extension of rainbow trout fillets by at least a factor of two. Bono and Badalucco [48] studied the effect of MAP (50% CO_2_/50% N_2_ in combination with ozonation (OZ, 0.3 mg L^−1^) on microbiological, biochemical, and sensory parameters of striped red mullet stored for 21 days at 1 °C. Samples stored in air were taken as the controls. The results revealed that (OZ) significantly retarded bacterial growth in fish muscle, with the TVC remaining below 6 log cfu/g until day 10 of storage. Chemical indices of MAP+OZ and MAP samples remained significantly lower than control samples throughout storage. Sensory evaluation indicated that both MAP and OZ+MAP treated samples were judged as acceptable during the first 10 days of storage. Kostaki et al. [49] evaluated the combined effect of MAP (M1 = 40% CO_2_/50% N_2_/10% O_2_; M2 = 60% CO_2_/30% N_2_/10% O_2_) and thyme oil (0.2% *v*/*w*) on the quality and shelf life of fresh filleted sea bass, during storage for 21 days at 4 °C. Controls (A) included aerobically packaged sea bass fillets. The pseudomonads and H_2_S-producing bacteria dominated the microflora of sea bass fillets, irrespective of treatment, while LAB were also part of the dominant microbial association. The TVC of controls exceeded 7 log cfu/g after 7 days, while treatments A+T, M1, M2 and M2+T reached the same value on days 9, 10, 12 and 19, respectively. As regards sensory evaluation, thyme oil improved product sensory acceptability when used in combination with M2, resulting in a shelf life of 17 days compared to 6 days for controls. Hassoun and Karoui [50] evaluated the quality of whiting fillets stored under MAP (MAP1: 50% N_2_/50% CO_2_ and MAP2: 80% N_2_/20% CO_2_) compared to counterparts stored aerobically (control samples) for 15 days at 4 °C. The results showed that the MAP1 treatment had a clear preservative effect on fish quality by decreasing the values of chemical quality indices and delaying the softening of fish flesh compared to controls. Silbande et al. [51] investigated the shelf life of yellowfin tuna from Martinique stored on ice (AIR-0 °C), VP (4 °C—first week and 8 °C—second week) and MAP (70% CO_2_/30% O_2_-4/8 °C). According to sensory evaluation results, the AIR treated tuna was rejected on day 13, when the TVC was 10^6^–10^7^ cfu/g. MAP, as well as VP, provided no extension of product shelf life. At the point of sensory rejection, *Brochothrix thermosphacta* and pseudomonads dominated the microbiota of the AIR products, while *B. thermosphacta* alone or in a mixture with *B. thermosphacta*, Enterobacteriaceae and LAB dominated the microbiota of MAP and VP samples, respectively. Lipid oxidation values were higher in MAP samples compared to AIR samples. Milne and Powell [52] studied the microbiota dynamics in fresh chilled Atlantic salmon fillets packaged under MAP. Skinless fillet portions were packaged in pouches containing 96% CO_2_ and stored for 38 days at <1 °C. The results showed an initial TVC of 10^2^ cfu/g on day 0. TVC reached 10^6^ cfu/g after 25 days and 10^8^ cfu/g after 31 days. On day 31, the microbial community was dominated by *Pseudomonas* spp., as determined by the identification of isolates and sequencing of a 16S rRNA gene clone library. *P. phosphoreum* (SSO), was not identified during the study. Alfaro and Hernandez [53] combined conventional methods and PCR sequencing to study the bacterial population dynamics during spoilage of Atlantic horse mackerel fillets packaged under MA (48% CO_2_/50% N_2_/2% O_2_) at 6 °C. The microflora was genetically characterized using 16S rRNA gene sequencing from isolates obtained from Long and Hammer agar. On the expiration date (day 5), the TVC on this selective medium was similar to the LAB counts on MRS agar. The PCR approach showed that *Photobacterium, Arthrobacter, Chryseobacterium* and *Pseudoclavibacter* spp. (44.5% of total) were the dominant microflora of the fish at the beginning of storage. The later stages of spoilage, however, were dominated by *Serratia, Shewanella* and *Yersinia* spp. (over 57.2% of the total). *Carnobacterium* spp. was the dominant species of LAB and was identified at both the beginning and end of storage. Fuentes et al. [54] studied the effect of NaCl replacement by KCl and packaging on the quality of smoked sea bass. Samples were salted with 100% NaCl or 50% NaCl-50% KCl, then smoked, packaged in air, VP, or MAP (70% O_2_/30% N_2_) and stored at 4 °C. Partial replacement of Na by K did not affect TVB-N, TMA-N, TBA, TVC, or sensory parameter values. The formation of BAs (histamine, putrescine, and cadaverine) was delayed by using the mixture of salts. Product shelf life was extended under VP and MAP in comparison to air. Noseda et al. [55] investigated the growth pattern and metabolite production of spoilage microorganisms of sushi catfish fillets packaged in air, VP and MAP (MAP 1: 50% CO_2_/50% N_2_ and MAP 2: 50% CO_2_/50% O_2_), stored at 4 °C. The shelf life of the fillets packaged in air was 7 days, in VP: 10 days, in MAP1: 12 days and in MAP2: 14 days. The dominant flora in air and VP fillets, identified by partial 16S rDNA sequencing at the end of the shelf life, consisted of Gram(−) bacteria (Serratia and Pseudomonas genera). In contrast, LAB (*C. maltaromaticum* and *C. divergens*) and *B. thermosphacta* dominated spoilage microflora in the MAP samples. Schirmer et al. [56] packaged fresh salmon under MAP (20% CO_2_) in a brine solution, to which various combinations of citric acid (3% *w*/*w*), acetic acid (1% *w*/*w*) and cinnamaldehyde (200 μg/mL) had been added. The combination of (CO_2_ + organic acids) completely inhibited bacterial growth for 14 days under refrigerated conditions. The individual addition of carbon dioxide, acetic acid and citric acid, inhibited TVC, LAB, Enterobacteriaceae and H_2_S reducing bacteria. Cinnamaldehyde did not affect bacterial growth. Van Haute et al. [57] studied the potential combination of MAP or VP and cinnamon essential oil (CEO) with the aim of prolonging the shelf life of salmon. Salmon was stored in VP or in MAP (60% CO_2_/40% N_2_) at 4 °C after dipping in a solution of 1% CEO. The results showed no effect of CEO on the microbial spoilage of VP or MAP. El-Sayed et al. [58] packaged chilled Atlantic salmon fillets under MAP with the addition of rosemary extract (RME). The authors reported notable antimicrobial activity of RME, resulting in a reduction in *Bacillus cereus/thuringiensis* and *Citrobacter freundii* counts, which dominated the microflora of the controls. 

#### 4.1.2. Use of Modified Atmosphere Packaging for the Preservation of Fishery Products

Gunsen et al. [59] determined the shelf life of marinated seafood salad packaged under MAP and the effect of gas mixtures M1 (70% CO_2_/30% N_2_), and M2 (50% CO_2_/50% N_2_) during storage at 2 °C. Results showed that seafood salad was acceptable after 7 months of storage compared to air-packaged marinated seafood salad, which reached the limit of acceptability after 4 months of storage. Mastromatteo et al. [60] investigated different packaging systems aimed at prolonging the shelf life of ready to eat peeled shrimp. A thymol coating (conc. 1000 ppm) in combination with MAP (5% O_2_/95% CO_2_) was used. The results showed that MAP affected the TVC (2 log reduction compared to air). When MAP was combined with the thymol coating, a 14 day shelf life was recorded compared to 5 days for the air packaged controls. Nirmal and Benjakul [61] studied the effect of MAP on the quality of Pacific white shrimp treated with the addition (or no addition) of green tea extract (GTE; 1 g/L) combined with or without ascorbic acid (AA; 0.05 g/L), stored under refrigeration for 10 days. Untreated shrimp stored under MAP had lower levels of psychrotrophic bacteria, Enterobacteriaceae and H_2_S-producing bacterial counts compared to shrimp stored in air (controls). MAP slightly reduced melanosis formation and somewhat improved sensory acceptability scores. Shrimp stored under MAP and treated with GTE showed less notable changes in chemical and microbiological indices as well as the lower formation of melanosis compared to both MAP treated samples and controls. MAP combined with GTE and AA showed the highest microbial growth inhibition and resulted in the highest product acceptability compared to the other treatments. Qian et al. [62] investigated the effect of immersion in quercetin (0.05 g/L), cinnamic acid (0.025 g/L) and 4-hexylresorcinol (0.025 g/L) solutions combined with MAP (80% CO_2_/10% O_2_/10% N_2_) on the quality of Pacific white shrimp during storage for 12 days at 4 °C. The preservatives helped to retard melanosis and protein breakdown of shrimp, but their inhibitory effect on TVC, TVB-N and K-value was limited. On day 12 of storage, the dipped and MAP samples were lighter in color and recorded lower TVC, K- and TVB-N values. Gornik et al. [63] used MAP (10% O_2_/80% CO_2_/10% N_2_) to package Norwegian lobster. The results showed that product shelf life was extended by 13 days when stored under MAP at 1 °C. The main SSO of MA packaged Norway lobster was *P. phosphoreum*. Ulusoy and Özden [64] studied the effect of MAP (MAP1: 50% N_2_/50% CO_2_ and MAP2: 100% CO_2_) on the sensory, chemical, and microbiological parameters of stuffed mussels stored at 4 °C. Based primarily on sensory evaluation, the shelf life of control (air packaged) samples was 11 days, while MAP1 and the MAP2 resulted in a shelf life of 13 days. Masniyom et al. [65] investigated the effect of MAP on the quality of green mussels stored at 4 °C. An increase in CO_2_ concentration resulted in proportional retardation of bacterial growth. The odor and flavor of MAP samples (addition of 80% and 100% CO_2_), was judged as acceptable throughout the 12 day storage period, compared to 6 days of the samples stored in air. Thus, MAP (80% CO_2_/10% O_2_/10% N_2_) was selected as the optimum condition to extend the shelf life of green mussels. 

Fall et al. [66] inoculated cooked and peeled shrimp with *Lactococcus piscium* CNCM I-4031 and investigated its antimicrobial effect against *B. thermosphacta*. The product was packaged under MA and stored at 8 °C. The control was spoiled by *B. thermosphacta* after 11 days, resulting in the development of strong butter/caramel off-odors. In the presence of *L. piscium*, a 10+ day product shelf life extension was achieved. The antimicrobial effect of *L. piscium* was related to a pH drop from 6.6 to 5.6.

#### 4.1.3. Use of Modified Atmosphere Packaging to Control Food Pathogens in Seafood

As regards the effect of MAP on food pathogens, concerns have been raised on the ability of psychrotrophic pathogens, i.e., *L. monocytogenes*, *A. hydrophila* and *Y. enterocolitica* to grow in MAP fishery products. According to the literature, the risks associated with foodborne pathogens in MAP are equal to those of foods stored aerobically. More specifically, Church [67] showed that the growth/survival of *L. monocytogenes*, *Aeromonas* spp., *Y. enterocolitica*, and *Salmonella typhimurium* in MAP cod and rainbow trout was similar to their aerobically stored counterparts. Provincial et al. [68] investigated the ability of *V. parahaemolyticus* and *A. hydrophila* inoculated onto sea bream fillets, packaged under MAP, to survive at 0 °C and 4 °C. The atmospheres used were MAP1: 60% CO_2_/40% N_2_, MAP2: 70% CO_2_/30% N_2_ and MAP3: 80% CO_2_/20% N_2._ Aerobically packaged samples were taken as the control. Head space gas analysis, microbial counts and confirmation tests of pathogens were monitored for 16 days. The results showed that all of the MAs used effectively reduced the microbial load of sea bream fillets compared to controls. Storage temperature was the main parameter affecting microbial proliferation. *V. parahaemolyticus* did not grow at both temperatures, 0 °C and 4 °C (except in controls), while *A. hydrophila* grew well at 4 °C but was inactivated at 0 °C. Newell et al. [69] conducted a challenge study to determine the possible formation of botulinum toxin in mussels. The product was inoculated with non-proteolytic *C. botulinum* spores and stored under MAP at 4 °C and 12 °C. A cocktail of six strains of non-proteolytic *C. botulinum* spores (10^3^ spores/mL) were introduced into live mussels by immersing the product in a seawater solution at 4 °C with cultured algae (10^4^ cells/mL). Mussels were packaged in (i) a 60–65% O_2_ MA and (ii) under vacuum conditions. Feeding cultured algae to live mussels for 6 h led to the uptake of spores into mussel tissue (500/g) and the mussel GI tract (100/g). Botulinum toxin was absent in VP and MAP mussels, even at the higher temperature (12 °C) for at least 12 days or at 4 °C for at least 21 days, which exceeds their normal shelf life. In contrast, in control samples of cooked mussels, the clostridium toxin was produced in the absence of oxygen. 

### 4.2. Vacuum Packaging 

Vacuum packaging (VP) refers to the technique in which the air is removed from a pack prior to sealing. Its principal purpose is to remove oxygen by bringing the packaging material into intimate contact with the product [70]. During the pulling of the vacuum, the concentration of oxygen in the package is drastically reduced by 97–99%, inhibiting the proliferation of aerobic SSO (specific spoilage organisms) responsible for product spoilage. The reduction in oxygen concentration also reduces the degree of oxidation. If, for example, the pressure is reduced from 1000 mbar (1 atm) to 100 mbar, an equivalent of around 2.1% oxygen will remain in the package.

The packaging material used for VP must provide a high gas and moisture barrier in addition to heat sealability [31]. Such barrier (PET/LDPE, PA/LDPE, LDPE/PA/LDPE, etc.) and high barrier materials (PET/PVdC/LDPE, PA/PVdC/LDPE, PA/EVOH/LDPE, LDPE/EVOH/LDPE, PS/EVOH/LDPE, PP/EVOH/PP, etc.), usually in the form of multilayer films or sheets, are suitable for the long-term storage of vacuum-packaged cooked muscle-based products. The oxygen permeability of the vacuum packaging film for cooked and processed meat/fish products is usually below 150 cm^3^/m^2^.24 h. atm in cases where PET or PA are used as barrier materials or lower than 10 cm^3^/m^2^.24 h. atm when PVdC or EVOH are used as high barrier materials.

Vacuum packaging also works very well for frozen fish and fishery products, where the exclusion of air helps to reduce freezer burn for fatty fish, such as salmon, mackerel, sardines, etc. A significant advantage of VP is that the pack volume is practically the same as the product volume, with no ‘empty’ space inside the pack. Compared to VP, MAP is a more versatile process, as the former relies solely on removing air, while the latter can be tailored to the particular foodstuff, with different gases and different concentrations of gases in the mixture used to provide the maximum shelf life for the particular product. For example, for packaging seafood, the concentration of CO_2_ and O_2_ in the MA can be varied depending on the fish fat content: oily fish require the absence of oxygen, while prawns benefit from the presence of oxygen.

A specific application of VP is sous vide processing, which originated in France in the 1970s. In this preservation technique, the food product is thermally pasteurized in a gas and moisture tight vacuum pouch or tray [71]. After pasteurization, the product is cooled to 4 °C, stored, distributed and kept under chilled conditions and reheated before consumption. The shelf life of sous vide-processed products varies, ranging from 7 to 90 days depending on the specific food being processed, the process used, and regulations governing product production. The shelf life of sous vide-processed Indian shrimp was extended up to 4 weeks, compared to 15 days for VP and 8 days for air-packed samples [72]. Sous vide processing of foods maintains the flavor compounds within the package, resulting in superior flavor and other sensory attributes, including increased tenderness, satisfactory color retention and minimal nutritional loss as nutrients do not leach out in cooking waters.

While VP functions by restricting the growth of SSO, the main safety issue with VP products is the enhanced probability of the formation of *C. botulinum* toxin before the product is rejected due to the deterioration of sensory attributes. This usually occurs at temperatures higher than those of refrigeration, which are favorable for the growth of *C. botulinum* and for sufficient time to allow toxin formation. Between the two groups *of C. botulinum* (proteolytic group and non-proteolytic group) the former includes *C. botulinum* type A and some of types B and F. The latter includes *C. botulinum* type E and some of types B and F. While the vegetative cells of all *C. botulinum* types are easily killed by heat (i.e., boiling for 10 min), the spores produced by *C. botulinum* are very resistant to heat. The heat resistance of spores of the proteolytic group is much higher than that of the nonproteolytic group. For type A and proteolytic types B and F, the minimum temperature for growth is 10 °C, while the respective temperature for the nonproteolytic *C. botulinum* type is 3.3 °C. As VP and MAP extend the shelf life of foods under refrigeration, there is more time for *C. botulinum* to grow and produce its toxin. With an increase in storage temperature, the time required for toxin formation is significantly reduced. According to the literature, temperatures of retail display cases range between 4–10 °C. Similar studies on home refrigerators have shown that temperatures can often exceed 10 °C [32].

Vacuum skin packaging (VSP) is a technique developed to overcome some of the disadvantages of the traditional VP and MAP (Figure 1). The VSP concept is based on a highly ductile plastic barrier film laminate (upper web), which, upon heating and the application of vacuum conditions, is draped over a food product placed on a rigid film or tray (lower web), thereby forming a second skin [71]. The upper web and the lower web are completely sealed to each other, enclosing the product entirely. VSP in the UK market includes sliced cooked and cured meats and fish products (i.e., peppered mackerel). In VSP the lower web consists of a rigid PVC or PP or PET tray, while the top film is made of LDPE or LLDPE lamination and/or co-extruded polymers, such as ionomers. In VSP, the adhesion of the plastic film to the food surface diminishes any headspace excluding oxygen and limits product juice exudation. This is of considerable importance because (i) any liquid within the package serves as a substrate for microbial proliferation, reducing product shelf life, and (ii) the presence of any liquid within the package renders the product unattractive to the consumer [73]. Finally, VSP is suitable for frozen food applications since the second synthetic skin prevents the formation of ice crystals on the product surface, resulting in the elimination of freezer burn and dehydration. Sliced products and smoked fish fillets are usually packaged using VSP.

The oxygen permeability of the top film varies depending on the type of product to be packaged, i.e., for fresh fish and meat, permeability may be desirable, while oxygen and moisture permeation should be prevented in dried fish and meat.

#### 4.2.1. Use of Vacuum Packaging for Fish Preservation

Masniyom et al. [75] investigated the effect of VP and MAP (60% CO_2_/10% O_2_/30% N_2_) on the keeping quality of tilapia fillets during storage at 4 °C. Maximum inhibition of the bacteria mesophilic and psychrotrophic growth was achieved under MAP. However, samples packaged under MAP exhibited an increase in exudate loss, suggesting muscle protein denaturation via the H_2_CO_3_ that was formed. Thiobarbituric acid reactive substances (TBARS) of samples kept under VP were lower than those stored under other conditions throughout the storage of fifteen days. The odor and flavor of samples under MAP, VP and air were acceptable for 15, 12 and 6 days, respectively. Erkan [76] investigated the biochemical, microbiological and sensory changes of “çiroz”, a Turkish salted dried fish product, during storage and packaged using an oxygen absorber and under vacuum conditions. Based on the TVB-N values and sensory evaluation, both active and VP samples, showed a shelf life of 6 months under refrigeration. Microbiological parameter values remained under the limiting values during storage in both packaging treatments. Comparison of the two packaging treatments led to the suggestion that active packaging may be an alternative to VP. Silbande et al. [77] carried out microbiological, chemical and organoleptic analyses on red drum fish fillets. The shelf life of whole iced fish was 15 days and that of VP and MAP fillets was 29 days. On these days, the TVC reached 10^7^–10^8^ cfu/g. At the point of sensory rejection, LAB was the dominant group in both types of packaging. TVB-N and TMA-N values were low and did not comprise reliable spoilage indicators. Wang et al. [78] investigated the effect of VP and MAP on microbial flora, physico-chemical parameters and sensory attributes of Pacific white shrimp stored for 14 days at −0.8 °C. The results showed that, in comparison to air packaging, VP, MAP1 (40% CO_2_/30% O_2_/30% N_2_) and MAP2 (75% CO_2_/10% O_2_/15% N_2_) more effectively retarded the growth of *Aeromonas* spp., *Pseudomonas* spp., H_2_S-producing bacteria and Enterobacteriaceae, while also reducing TVB-N and TMA-N contents. According to sensory scores, the shelf life of fresh Pacific white shrimp was 4 days for samples packaged aerobically, 7 days under VP, 9 days under MAP1 and 11–12 days under MAP2. Tsogas et al. [79] evaluated the effect of light salting in combination with VP on the keeping quality of mullet fillets during refrigerated or frozen/refrigerated storage, stored for up to 5 weeks. Treatments included: (a) salting of fish for 24 h, aerobic packaging and storage at 4 °C (C); (b) salting for 24 h, VP and storage at 4 °C (VP1); (c) salting for 12 h, VP and storage at 4 °C (VP2); (d) salting for 24 h, VP, freezing for 9 days, defrosting and storage at 4 °C (FVP1). LAB were found to dominate the spoilage microflora of fish. Based on microbiological results, VP1 was the most effective treatment for extending the product shelf life. Sensory analysis suggested a product shelf life of ca. 8–9, 10, 13, and 27–30 days for C, VP2, FVP1 and VP1 samples, respectively. Ramírez-Suárez et al. [80] determined the effect of adding a natural antimicrobial agent (P ronat) to vacuum packaged frankfurters, made from jumbo squid muscle, on product shelf life. Frankfurters with and without P ronat (0.1%) were vacuum packaged and stored at 2–4 °C for up to 28 days. The samples containing P ronat maintained good condition up to 21 days. The main changes recorded were attributed to color, WHC and drip loss. The P ronat concentration used in squid frankfurters greatly improved their microbiological shelf life but did not considerably improve the physicochemical quality indicators assessed in the present study.

Özogul et al. [81] studied the effect of rosemary and sage tea extracts on VP sardine fillets stored at 3 °C for 20 days. The addition of both extracts showed lower levels of NH_3_ and biogenic amine accumulation in sardine fillets. After 20 of storage, the Putrescine and Cadaverine contents of treated samples were 100-fold lower than those of control groups. Houicher et al. [82] evaluated the preservation potential of vacuum packaged sardine fillets with the addition of *Mentha spicata* and *Artemisia campestris* ethanolic extracts. Fillets were stored at 3 °C for 3 weeks. The groups tested included: (i) the control samples; (ii) VM (samples treated with 1% mint extract); (iii) VA (samples treated with 1% artemisia extract). Control samples had a shelf life of 10 days, whereas the combined treatment had a shelf life of 17 days. The addition of individual extracts resulted in shelf lives between 10 and 17 days. Besides shelf life extension, the combined use of VP and natural extracts also retarded sardine lipid oxidation. Attouchi and Sadok [83] investigated the effect of laurel and/or cumin EOs in fresh VP wild and farmed sea bream fillets. The study showed that treatments with both EOs exhibited reduced TVC by ca. 0.5 to 1 log cfu/g and reduced fat oxidation by ca. 40% than controls, resulting in a 5 day shelf life extension of iced stored sea bream fillets.

Ibrahim and Vesterlund [84] inoculated vacuum packaged raw Atlantic salmon with 16 selected lactic acid bacteria and evaluated their inhibitory properties against 32 spoilage organisms. Their results showed that *Lactococcus lactis* subsp. *Lactis* provided the strongest inhibition, prolonging product shelf life by 3 days in comparison to controls. It was also demonstrated that the addition of *L. lactis* did not cause changes in the sensory and textural properties of the salmon samples. Speranza et al. [85] inoculated marinated anchovies with two probiotic strains (*Lactobacillus plantarum* and *Bidifobacterium animalis* subsp. *lactis*). Samples were then aerobically packaged or VP (in oil and in a diluted brine) and stored at 4 °C for up 21 days. Sensory scores proved to be the limiting factor for the determination of product shelf life. The most acceptable sample for the entire 3 week storage period was that which was packaged in diluted brine. Leroi et al. [86] used selected LAB strains (*L. piscium* EU2241, *Leuconostoc gelidum* EU2247, *Lactobacillus sakei* EU2885 and *Staphylococcus equorum* S030674) to preserve VP cold smoked salmon (CSS), stored at 8 °C. The population of SSO, *P. phosphoreum*, *B. thermosphacta*, *C. divergens* and *Serratia proteamaculans* was monitored with time. Each SSO was affected differently by LAB strains with no correlation being established between the SSO inhibition, sensory data and the acidification caused by the LAB cultures. Duan et al. [87] packaged lingcod fillets under vacuum conditions or MA after vacuum-impregnating fillets with chitosan (Ch) solutions containing krill oil (KO). Treated fillets were then refrigerated at 2 °C for up to 21 days. The combined treatment of either VP or MAP with chitosan resulted in a reduction in key fish quality parameter values such as TVC, TVB-N and TBARS. The Ch-KO coating did not change the color of the fillets or affect consumer acceptability of both raw and cooked filleted samples. Sensory analysis showed that Ch-coated, cooked lingcod samples recorded higher color, odor and texture scores compared to control samples. Matějková et al. [88] combined VP with high hydrostatic pressure (HHP) as a means of preservation of trout stored at 3.5 °C. The biogenic amines content, monitored during fish storage decreased in the VP+HHP treated trout, resulting in a shelf life extension from 5–6 days for the control sample to 3–4 weeks for the VP+HHP treated fish. Günlü et al. [89] investigated the possible synergistic effect of VP and HHP in retarding the spoilage of rainbow trout fillets. The combined use of VP+HHP resulted in a 4 day shelf life extension for trout fillets stored under refrigeration. Navarro-Segura et al. [90] developed and evaluated a pilot plant packaging system based on oregano essential oil (0.1% OEO) vapors (67 μL L^−1^), high vacuum (5–10 hPa) prior to MAP to extend the shelf life of farmed sea bream (treatment T3). The results were compared to conventional dipping of whole fish in 0.1% OEO (treatment T1) and filleted fish (treatment T2). Results revealed that T3/T2 samples showed the largest reduction in microbial growth after storage for 28 days at 4 °C, with a ratio of 1/2.6 log units for *Enterobacteria*/LAB compared to untreated samples. The TMA increased from 2.6 mg kg^−1^ in T1 and T2/T3 samples to 11.2 and 8.6/9.6 mg kg^−1^, respectively, after 28 days.T3/T2 samples had a shelf life of at least 28 days but only 7 and 21 days for the untreated andT1 fillets, respectively. Chan et al. [91] investigated the effect of MAP (CO_2_/N_2_, 60%/40%) and VSP of fresh Atlantic salmon fillets that had been kept at 0 °C for 6–8 days after slaughter. Fillets were stored under refrigeration for 3 weeks. Both MAP and VSP gave similar values for drip loss, texture and microbiological shelf life. Fillets kept in MAP had consistently lower pH values with a darker, more reddish, and yellowish color than VSP fillets. The microbiological shelf life (limit of 10^6^ cfu/g) of MAP and VSP fillets was ca. 18–20 days, while that of air stored fillets was <13 days. Dordevic et al. [92] determined the biogenic amine content of escolar fish fillets packaged under vacuum conditions (control) and VSP, stored for 9 days at 2 °C. Results showed that the initial total value of biogenic amines (BAs) was low but increased considerably after 9 days of storage. VSP can be recommended for storage at 2 °C for a period of 7 days (total BA: 21.31 mg/kg), but not for 9 days, due to high histamine content (total BA content: 376.96 mg/kg). Finally, Esteves et al. [93] studied the effect of air packaging, VP and MAP on physicochemical, microbiological and organoleptic quality parameters as well as on the shelf life of gray triggerfish fillets stored for 2 weeks under refrigeration. VP retarded and MAP inhibited TVB-N of fillets in comparison to air packaged fillets. TVC and psychrothropic bacteria of fillets in aerobically packaged samples reached the limit value of 7 log (cfu/g) on day 10, while the TVC for VP and MAP samples remained below this limit throughout storage. Based on all analytical parameters determined, but primarily on sensory attributes, the shelf life of fillets packed in air was 8, 15 and 12 days for aerobically packaged, VP and MAP samples, respectively.

#### 4.2.2. Use of Vacuum Packaging for the Preservation of Fishery Products

Atrea et al. [94] investigated the effect of VP in combination with oregano EO (OEO) for the preservation of Mediterranean octopus stored for 23 days under refrigeration. VP+ EO treated octopus samples showed significantly lower TVB-N and TMA values in comparison to controls. Sensory evaluation along with microbiological analysis showed that increasing the concentration of OEO from 0.2 to 0.4% increased the octopus shelf life from 11 to 20 days. Matamoros et al. [95] inoculated batches of cooked, peeled shrimp with seven LAB strains separately, with the purpose of extending the product shelf life. LAB strains were isolated from seafood products. The initial concentration of inocula was 5 log cfu/g. Seafood products were then VP and stored for up to 4 weeks at 8 °C. Spoilage was determined by organoleptic evaluation after 1 and 4 weeks. Four of the seven LAB strains used (two *L. gelidum* and two *L. piscium* strains), showing the best results, were used in an identical experiment involving cold-smoked salmon. In this second experiment, *L. piscium* strains exhibited the highest inhibition capacities, providing a product shelf life of 28 days.

### 4.3. Active Packaging

European Commission Regulation (EC) No 450/2009 [96] defines active packaging (AP) as that which provides functions beyond the inert barrier and traditional protection from the external environment. Likewise, AP can also be defined as the type of packaging in which interaction occurs between the package, the product and the environment, with the aim of extending the product shelf life or enhancing the safety or sensory attributes, while maintaining product quality [97,98]. The basic principle of AP is the addition of certain active compounds into the packaging material or inside the packaging container which have a specific function, i.e., antioxidant, antimicrobial function, reduction in cholesterol content, etc., with the general objective of maintaining product quality and safety, while extending product shelf life. AP technology is based either on scavenging or emitting systems which are added to the package either to remove (e.g., O_2_, CO_2_, moisture, odors, ethylene) gases or to emit (i.e., CO_2_, ethanol, flavors, antimicrobials, antioxidants) during product storage and distribution. Active compounds are either (i) contained in individual sachets or labels placed within or on the packaging material, (ii) coated on the internal surface of the container or (iii) directly incorporated into the packaging material, i.e., through extrusion or lamination. The main AP applications for fishery products include O_2_ scavengers, moisture absorbers, CO_2_ emitters, antimicrobial and antioxidant release systems [99,100] (Table 1). These will be discussed in more detail in the following sections.

#### 4.3.1. O_2_-Scavengers

As previously mentioned, fish and seafood products are particularly susceptible to oxygen, which leads to (i) the growth of aerobic microorganisms (i.e., pseudomonads), (ii) oxidation, which causes undesirable color changes (i.e., discoloration of myoglobin), off-odors and flavors (i.e., rancidity due to fat oxidation), and (iii) the degradation of nutrients (i.e., oxidation of vitamin E, β-carotene, ascorbic acid), which negatively affects food quality. Thus, all the above deteriorative and spoilage reactions in seafood can be controlled by the availability of oxygen within the food package. Even though seafood prone to oxidation can be substantially protected using MAP or VP, these technologies do not completely remove O_2_. Usually, during the creation of a vacuum, 0.3–3% of air remains within the package. Furthermore, the O_2_ permeating through the packaging material (i.e., film) cannot be removed by using MAP or VP. An O_2_-scavenger absorbs the residual O_2_ within the package at any point in time, creating an atmosphere containing ≤ 0.01% O_2_ which minimizes the quality changes of oxygen-sensitive foods. Mitsubishi Gas Chemical Company was the first to commercialize oxygen absorbers by the trade name Ageless^®^ in the late 1970s. Oxygen absorbers are used to prevent discoloration of fresh and cured fish, oxidative flavor changes and mold spoilage of intermediate and high moisture seafood products. Most oxygen scavenging concepts are based on (i) iron powder oxidation:2Fe → 2Fe^2+^ + 4e^−^(1)
O_2_ + 2H_2_O + 4e^−^ → 4OH^−^(2)
2Fe^2+^ + 4OH^−^ → 2Fe (OH)_2_(3)
2Fe (OH)_2_ + H_2_O + O → 2Fe(OH)_3_(4)

In this case, the oxygen scavenging component can be added to a package in the form of a sachet, sticker, coating or direct incorporation into the packaging film (Figure 2). They can also be based on (ii) vitamin C oxidation, (iii) unsaturated fatty acid oxidation, (iv) photosensitive dye oxidation, and (v) enzymic oxidation (i.e., through the use of glucose oxidase and alcohol oxidase), etc. [100].

Oxygen scavengers have been more widely used in Japan, Australia and the USA compared to Europe. This trend may be related to (i) the strict EU legislation as regards food-contact materials, which cannot always follow innovations in packaging technology, and thus, currently prohibit the application of many of these materials and (ii) the limited knowledge and demonstration of their effectiveness and safety as shown by independent research organizations. Commercial O_2_ scavengers include: Ageless^®^ (Mitsubishi Gas Chemical, Tokyo, Japan), ActiTUF^™^ (M&G Finanziaria s.r.l. Tortona, Italy), Cryovac^®^ OS Film (Sealed Air Corporation, Charlotte, NC, USA), ATCO^®^ (Standa Industrie, Caen, Italy), FreshMax^®^ (Multisorb Technologies, Buffalo, NY, USA), Freshilizer^®^ (Toppan Printing Co., Tokyo, Japan), Oxyguard^TM^ (Toyo Seikan Kaisha, Ltd., Tokyo, Japan), Bioka Oxygen Absorber Sachets (Bioka Ltd., Kantvik, Finland), etc.

#### 4.3.2. CO_2_-Emitters

The method of preserving food products using CO_2_ has been extensively used in MAP. High CO_2_-levels (10% to 80%) are used in foods of high moisture content, such as seafood and meat products for the inhibition of surface microbial growth, resulting in product shelf life extension. CO_2_ has a significant effect on the growth of Gram(−) bacteria such as the pseudomonads.

A carbon dioxide emitting system functions in a complimentary manner to MAP, overcoming its drawbacks, i.e., dissolution of CO_2_ in the food aqueous phase during storage. Such systems are based on NaHCO_3_, FeCO_3_, ascorbic acid, citric acid, etc. In the presence of sufficient moisture, sodium bicarbonate, when used in combination with ascorbic acid or citric acid, generates CO_2_. This is a simple and economical technique requiring no expensive equipment and pure gases [100]. Commercial CO_2_ emitters include: CO_2_^®^Fresh Pads (CO_2_ Technologies Inc., Urbandale, IA, USA), UltraZap^®^ Xtenda Pak pads (NOVIPAX, Oak Brook, IL, USA), SUPERFRESH (vdP International, Ellewoutsdijk, The Netherlands), etc. (Figure 3a).

#### 4.3.3. Moisture Regulators

Seafood has a high vapor pressure, and hence, this results in high relative humidity values in the container when such foods are packaged. In such environments, moisture will be trapped within the package due to (i) temperature fluctuations or (2) tissue drip loss from cut fish and fish products. Such excess moisture will cause microbial spoilage and/or condensation on the inner surface of the packaging material resulting in reduced consumer appeal. Effective control of excess moisture accumulation within a seafood package is achieved through the use of (i) a film material with specific water vapor transmission rate and (ii) a moisture absorber, such as silica gel, CaCl_2_, natural clays, modified starch, etc. Drip-absorbent sheets for liquid water control in high moisture fresh fish and shellfish consist of an absorbent polymer sandwiched between two paper-based layers. Such absorbent polymers include polyacrylate salts and graft copolymers of starch. For dried fish applications, natural clays (e.g., montmorillonite), silica gel, molecular sieves and CaO packed in sachets are used as desiccants [100]. Commercial moisture scavengers include: Dri-Loc^®^ (Sealed Air Co., Charlotte, NC, USA), MoistCatchTM (Kyoto Printing Co. Ltd., Kyoto, Japan), Fresh-R-Pax^®^ (Maxwell Chase Tech., Atlanta, GA, USA), etc. Showa Denko Co. (Minato-Ku, Tokio, Japan) developed a multilayer plastic film consisting of a layer of humectant propylene glycol sandwiched between layers of PVA. This moisture absorbing system can extend the shelf life of fresh fish by 2–4 days [102] (Figure 3b).

#### 4.3.4. Antimicrobial Packaging

Conventional methods for the inhibition of microbial growth in seafood include: refrigeration, freezing, thermal processing, smoking, salting, and drying, while more innovative methods include: high hydrostatic pressure, irradiation, ozonation, retort pouch processing, MAP, VP and the use of additives with antimicrobial properties. However, a number of the above technologies cannot be applied to fresh seafood products as they lose their ‘fresh’ character. Thus, research has more recently focused on the development of antimicrobial packaging (AmP) for the preservation of seafood and meat. Treatment of fish or meat with antimicrobial agents in the form of spraying, dipping or antimicrobial films enhances product safety and delays spoilage, as cross-contamination occurs primarily on the fish/meat surface due to improper handling and distribution. Antimicrobial films function by releasing antimicrobial agents on the fish surface, thus, both increasing the lag phase and retarding microbial proliferation, extending product shelf life, while maintaining product quality and safety. The classes of antimicrobials include: organic acids (benzoic, propionic, sorbic acid), bacteriocins, i.e., nisin, polysaccharides such as chitosan, enzymes (lysozyme, peroxidase, glucose oxidase), EOs (thyme, oregano, rosemary), herbal extracts, chelators (EDTA), acid anhydrides (SO_2_, ClO_2_). ZnO, TiO_2_, MgO and silver zeolite have been extensively studied among inorganic nanoparticles for antimicrobial packaging applications [99].

Antimicrobial films may also be prepared from natural macromolecules which have inherent antimicrobial properties. A typical example of this case is chitosan which also has good film-forming properties. Commercial antimicrobial packaging materials in the form of antibacterial and antifungal sheets, labels and films, silver-based trays and films include: Agion^®^ (AgIon technologies, Wakefield, MA, USA), Food-touch^®^ (MicrobeGuard Corp., Elk Grove Village, IL, USA), etc. Silver substituted zeolite is widely used in Japan as a food contacting layer of a laminated polymeric structure (Figure 4).

#### 4.3.5. Antioxidant Packaging

Antioxidants, mostly synthetic (BHA, BHT, PG, TBHQ), are widely used as food preservatives to prolong the shelf life of fatty foods, protecting them from oxidation. This especially applies to fatty foods containing unsaturated fatty acids, such as seafood products. Antioxidants have also been incorporated into plastic films (i.e., polyethylene, polypropylene) to prevent the polymer from degradation. Incorporation of BHT into the packaging films as an antioxidant has been commercially practiced for decades. However, the use of synthetic antioxidants has been recently limited due to the possibility of causing adverse effects on human health, even though they are very effective and stable [105]. For this reason, natural antioxidants such as vitamins C and E have been used to replace synthetic antioxidants in food and food packaging applications. Vitamin E maintains its stability under commercial processing conditions and is highly soluble in polyolefins. Besides vitamins, natural antioxidants in the form of extracts derived from plant tissues (citrus extract, grape seed extract, etc.) or EOs are being tested for the development of antioxidant packaging [106].

#### 4.3.6. Active Packaging Systems with Multiple Functionality

In applying the ‘active packaging’ principle for the effective shelf life extension of seafood, multiple function active systems are often used. Examples of this include the combination of a moisture absorber with a carbon dioxide emitter (Figure 3b) or an antimicrobial releasing system with a carbon dioxide emitter (Figure 3c). Such systems will significantly improve the storage stability of packaged seafood and extend the shelf life of products. Another typical example of multiple function active systems includes the synchronous use of an oxygen absorber and a CO_2_ emitter. When an O_2_ scavenger is used alone, as O_2_ is removed from the package a partial vacuum is created, leading to the collapse of the flexible package. To avoid this phenomenon, the O_2_ absorber is combined with a CO_2_ emitter, with the latter inhibiting surface microbial growth, resulting in extension of the product shelf life.

#### 4.3.7. Use of Active Packaging to Seafood Products Preservation

Mohan et al. [107] evaluated the quality of seer fish steaks packaged with and without an O_2_ scavenger using chemical and sensory analysis. Data showed that the O_2_ scavenger resulted in a reduction in the O_2_ concentration within the package to <0.01% after 24 h of storage. The shelf life of aerobically packaged samples was 12 days versus 20 days for O_2_ scavenger packages. At the point of sensory rejection, TVB-N and TMA-N levels were somewhat higher for aerobically packaged samples compared to their O_2_ scavenger counterparts. The rate of decrease for inosine monophosphate (IMP) and increase for hypoxanthine (Hx) was higher for aerobically packaged samples compared to their O_2_ scavenger counterparts. Mohan et al. [108] investigated the effect of an O_2_ scavenger on the formation of BAs in the same seer fish steaks during chilled storage. The BA content increased significantly in aerobically packaged samples in comparison to their O_2_ scavenger counterparts. Histamine, putrescine and cadaverine recorded high concentrations of 6.8, 14.6 and 14.7 μg/g, respectively, in air packaged samples on day 15 of storage. Spermidine and spermine concentrations increased slightly in the initial stages of storage and subsequently stabilized. Agmantine was not identified in the O_2_ scavenger packs during the entire storage period, whereas tyramine was determined only during the later stages of storage. López-De-Dicastillo et al. [109] developed active antioxidant food packaging films by incorporating ascorbic acid, ferulic acid, quercetin, and green tea extract (GTE) into an ethylene vinyl alcohol copolymer (EVOH) matrix. The antioxidant activity of the films was determined when in contact with brined sardines. The evolution of PV and MDA showed that test films retarded product lipid oxidation. Films containing GTE resulted in the best protection of sardines against lipid oxidation. Pereira de Abreu et al. [110] prepared an active food packaging film by coating conventional polyethylene films with a phenolic extract originating from barley husks (PEBH). Frozen Atlantic salmon was packaged in the test films. The results showed that PV, conjugated dienes (CD), conjugated triene hydroperoxides (TH), FFA, the totox value (TV), TBARS and the *p*-anisidine value (AV) were lower in films containing (PEBH), resulting in increased oxidative stability of salmon flesh. Zeid et al. [106] prepared antioxidant packaging films by incorporating thyme, rosemary, and oregano EOs into polylactic acid resin (PLA). The radical scavenging activity (RSA) of pure EOs ranged between 84.57% and 87.92% at a C = 10,000 mg/L (ppm) using the DPPH method. The film methanolic extracts recorded an antioxidant activity value 4–6% lower than pure EOs at the same concentration. Minced fish packaged in active films, showed a significant reduction in the degree of oxidation, ranging from 5% to 40%, on day 4 of storage in comparison to control films. Mexis et al. [111] evaluated the effect of an O_2_ absorber in combination with OEO on shelf life extension of Greek cod roe paste stored at refrigeration temperatures. Results showed that TVC reached the limit value of 7 log cfu/g on day 12–13 of storage for aerobically packaged samples and day 31–32 for samples containing OEO. Samples containing either the O_2_ absorber alone or the combination of O_2_ absorber + OEO remained lower than 7 log cfu/g throughout the entire 60 days of storage. At the point of sensory rejection, TBA (expressed as malondialdehyde) values reached 3.4mg/kg for samples containing the OEO, 3.2 mg/kg for samples containing the O_2_ absorber and 2.9 mg/kg for samples containing the O_2_ absorber + EO. The sensory shelf life was 24, 32, 60 and 60+ days for the control samples, samples containing oregano EO, samples containing the O_2_ absorber and samples containing the O_2_ absorber plus oregano EO. Neetoo and Mahomoodally [112] evaluated the antilisterial effectiveness of cellulose-based coated LDPE films containing nisin (Nis) and sodium lactate (SL), sodium diacetate (SD), potassium sorbate (PS), and/or sodium benzoate (SB) in binary or ternary mixtures on cold smoked salmon (CSS). The combination of nisin along with PS and SB exhibited the highest antilisterial activity, resulting in a reduction in the inoculated pathogen by up to 3.3 log cfu/cm^2^ compared to controls, after 10 days of storage at 21 °C. Hansen et al. [113] packaged salmon fillets under MAP, MAP plus a CO_2_ emitter and VP. Samples were stored at 1.2 °C for 25 days. MA was more effective at reducing bacterial growth compared to VP. Undesirable odors and drip loss were detected after 8 days for VP samples versus 15 days for MAP samples. The study concluded that MA packaging had a better preservative effect compared to VP. MAP, along with the CO_2_ emitter, gave better results compared to MAP alone; thus, CO_2_ emitters are suitable for the preservation of MA packaged farmed salmon fillet slices. Alboofetileh et al. [114] incorporated marjoram (MEO), clove (CLEO) and cinnamon essential oil (CEO) in alginate/clay nanocomposite films and evaluated their antilisterial effectiveness in a model solid food system stored at 10 °C for 12 days. The results showed that the films containing MEO exhibited a higher antilisterial effect. In a similar experiment, trout slices were inoculated with *L. monocytogenes*, packaged in an alginate-clay film containing MEO and stored for 15 days at 4 °C. The alginate-clay films containing MEO significantly delayed the growth of *L. monocytogenes* reaching final counts of 6.2 log cfu/g in comparison to 7.4 log cfu/g for control samples. Furthermore, the TVC, psychrotrophic count and TVB-N of fish packaged in active films were significantly lower in comparison to controls. Jalali et al. [115] investigated the effect of alginate/carboxyl methylcellulose composite coating (C-A) containing 1% and 1.5% clove EO (CEO) on the preservation of silver carp fillet stored at 4 °C for 16 days. A secondary objective of the study was to investigate the effect of these treatments to control *Escherchia coli* O157:H7 after inoculation in silver carp fillets. The results showed that C-A + CEO 1.5% resulted in the lowest biochemical and microbiological parameter values and the best sensory characteristics for the entire storage period. This treatment also decreased the population of *E. coli* O157:H7 to less than 2 log cfu/g beginning at day 4 of storage. Tokur et al. [116] coated whole trout with a whey protein coating containing thyme EO (TEO) held under refrigeration. The results of the study showed that an increase in the amount of TEO used resulted in an increase in the shelf life of the trout. Duan et al. [87] coated fresh lingcod fillets with chitosan solutions containing krill oil (KO) with or without the addition of cinnamon leaf EO (CLEO). Fillets were then VP or MAP and stored at 2 °C for up to 3 weeks. The chitosan coating, in combination with either VP or MAP, resulted in a reduction in: lipid oxidation, as reflected by the TBARS; chemical spoilage, as reflected by TVB-N; microbial spoilage, as shown by the TVC (2–4 log cfu/g reductions during storage). Acceptability sensory testing indicated preference of chitosan-coated, cooked lingcod samples compared to control samples, based on their superior texture and aroma. Shakila et al. [117] developed bone gelatin films containing chitosan (GC), clove (GL), and pepper (GP) EOs. All the films inhibited *Staphylococcus aureus*, *A. hydrophila*, and *L. monocytogenes*. Vacuum packaged fish steaks coated with GC and GL films resulted in a four day shelf life extension during storage at 4 °C. Pereira De Abreu et al. [118] studied lipid hydrolysis and oxidation in frozen Atlantic halibut packaged in an LDPE film containing barley husk extracts used as natural antioxidants. After 12 months, the FFA values in the samples packaged in the antioxidant film were similar to the FFA value in the controls after 9 months of storage. Comparable PV values in samples packaged in the antioxidant films were recorded one month later than those recorded in the control sample. After 6 months of storage, the concentration of TBARS expressed as malondialdehyde in the control sample was ca. 30–50% higher than that in samples packaged in the antioxidant film. Pereira De Abreu et al. [119] investigated the potential of antioxidant packaging to slow down the oxidation of PUFA in hake fillets. Samples were packaged in LDPE films containing natural antioxidants derived from barley husks (sample C1 = 7 mg/dm^2^ and sample C2 = 24 mg/dm^2^) or without the antioxidant (control sample). Samples were kept frozen for 12 months at −20 °C. After 6 months of storage, the TBARS levels were lower in samples packaged in both antioxidant films (C1 and C2) compared to control samples. Likewise, after 12 months, the TBARS values of C1 and C2 were lower (16% and 21%, respectively) than controls. Anisidine values in both C1 and C2, after 9 months of storage, were lower than those of controls after 12 months of storage. Remya et al. [120] prepared antimicrobial packaging films from chitosan (Ch), containing 0.3% *v*/*v* ginger EO (GEO). CH-GEO films were used to package barracuda fish steaks stored at 2 °C for 20 days. The results showed that the TVC and TVB-N of fish steaks packaged in Ch-GEO films were significantly lower than both the unwrapped controls and aerobically packaged fish steaks in a multilayer high barrier film. Based on sensory analysis, samples wrapped in the Ch-GEO film achieved a shelf life of 20 days versus 12 days for unwrapped controls and samples packaged in the high barrier film. In a similar study, Remya et al. [121] combined an O_2_ scavenger (OS) with an antimicrobial film (AM) for shelf life extension of fresh cobia fish steaks stored at 2 °C in a multilayer high barrier pouch. TVB-N and TBA were significantly reduced in fish steaks wrapped with the antimicrobial film containing the O_2_ scavenger (OS+AM). The TVC recorded a lag phase of 5 days for OS+AM samples compared to controls. Based on sensory evaluation, OS+AM samples had a shelf life of 30 days, while controls had a shelf life of only 15 days. Mexis et al. [122] evaluated the effect of an O_2_ absorber in combination with oregano EO on keeping quality of rainbow trout fillets stored at 4 °C. The results showed that in all cases when the O_2_ absorber was combined with OEO, a more pronounced inhibition effect resulted on *Pseudomonas* spp., Enterobacteriaceae and LAB. Based on sensory evaluation and microbiological data, control samples had a shelf life of 4 days, samples containing OEO (7–8 days), samples containing the O_2_ absorber (13–14 days) and samples containing the O_2_ absorber + OEO (17 days).

### 4.4. Intelligent Packaging

Intelligent packaging (IP) can be defined as ‘packaging that contains an external or internal indicator providing information as regards the history of the package and/or the quality of the food’ [31]. Intelligent packages are integrated with a target-specific sensor, which can store information on attributes such as freshness, gas leakage, microbial contamination, etc., and convey this information to the consumer. Sensors can be placed either outside the package measuring external environmental conditions or within the package measuring the quality of the packaged foodstuff. Such sensors should be easily activated and provide irreversibly changes having occurred. Therefore, IP can provide precise information about the ‘actual’ shelf life and directly indicate the current quality state of the food to ensure product quality and safety [123,124]. IP primarily includes (1) freshness indicators, (2) time temperature indicators and (3) leakage indicators.

#### 4.4.1. Freshness Indicators

Freshness indicators in the form of stickers or labels provide indirect information on product quality as a result of either microbial proliferation or biochemical changes occurring in a packaged food product. Such information may include (i) deviation from normal storage temperature, (ii) changes in gas concentrations, i.e., CO_2_, O_2_, NH_3_, H_2_S, diacetyl, etc. within the package, which indicate product quality deterioration. Microbiological quality can be evaluated through the determination of specific metabolites formed as a result of microbial growth and their reaction with specific indicators included within the package. In turn, deteriorative biochemical changes occurring in fish during storage, i.e., production of metabolites such as TVB-N, TMA-N, NH_3_ or BAs (histamine, putrescine, tyramine and cadaverine, etc.) provide the basis for the development of freshness indicators. BAs are non-volatile compounds and, thus, cannot be detected through sensory analysis. Thus, the development of an effective indicator of the presence of BAs would be very useful for the food industry. In contrast, H_2_S, a breakdown product of the amino acid cysteine, with an intense ‘rotten egg’ odor is formed during seafood spoilage by numerous bacterial spp. When bound to myoglobin, hydrogen sulfide forms a green pigment (sulfmyoglobin) which can be used for the development of a freshness indicator in red meat fishes. Normally, freshness indicators are attached to the packaging material, which react with volatile amines or other indicator compounds produced during the deterioration of fish and other seafood, and the degree of freshness/spoilage is usually indicated by a color change as a result of a chemical or enzymic reaction [73,100]. A number of fish freshness indicators have been described in the literature based on pH change [125]. As the packaged fish product begins to spoil, pH increases over time within the package headspace, which can be detected with an appropriate pH indicating sensor. The key element of such a sensor, i.e., a pH sensitive dye, will change color when placed in an alkaline environment. This occurs because when fish spoils, it releases metabolites of alkaline reaction, i.e., trimethylamine, dimethylamine, ammonia (known as TVB-N), which are detectable with appropriate pH indicating sensors. Such sensors can be prepared by the entrapment of a pH sensitive dye (i.e., bromocresol green) within a polymer matrix that responds, through visible color changes to the spoilage volatile metabolites [126,127]. The response of such sensors has been found to correlate with bacterial growth patterns (TVC and pseudomonads count) in cod and whiting fish samples, permitting ‘real-time’ monitoring of spoilage [128]. A variety of different freshness indicators have been reported in the literature for, i.e., CO_2_, amines, ammonia, ethanol and H_2_S [125].

#### 4.4.2. Time–Temperature Indicators (TTI)

Usually, TTIs are small labels or stickers that monitor the time–temperature history of a highly perishable product, such as fish exposed to abuse temperatures spanning from the production to the retail outlet or the end-consumer use. The use of TTIs in fish and seafood products is of paramount importance given the need to maintain product quality and safety. The basic idea behind a TTI is that food quality deterioration is more rapid at high, abuse temperatures due to the acceleration of microbial growth and biochemical reactions. These, in turn, cause specific changes in chemical (polymerization), electrochemical, enzymic, microbiological (based on reactions in crystal phase) or physical (melting, diffusion of a substance) parameters, which are usually expressed as a specific response, i.e., change in color, or a mechanical deformation. Besides visual indicators, radiofrequency identification tags (RFID) are also used for the same purpose. A RFID tag serves, basically, as an advanced data carrier. A reader that emits radio waves, collects data from a RFID tag, which are transmitted to a computer that processes the data. A RFID tag accomplishes this with the aid of a microchip attached to an antenna [100]. TTIs should:be activated in a simple way;provide a measurable change, which is a function of time and temperature;provide a short and irreversible response;correlate well with the degree of food deterioration.

Numerous TTIs have been proposed to date, however, very few have found commercial applications [129]. The CheckPoint^®^ TTI (VITSAB A.B., Limhamn, Sweden) is an enzymic TTI, based on a color change via pH decrease. The Keep-it^®^ indicator (Keep-it Technologies^®^ AS, Oslo, Norway) is based on a time-temperature dependent migration of a pH modifying agent. The 3M Monitor Mark^®^ (3M Co., St. Paul, MN, USA) is a diffusion polymer based indicator. The FreshStrips (FreshStrips B.V., Eindhoven, The Netherlands) based on shape memory of liquid crystals, etc.

#### 4.4.3. Leakage Indicators

Leak indicators are devices that, when attached to a package, provide information on the structural integrity of the packaging container throughout handling and distribution. For example, in a MAP application consisting of a high CO_2_ and low O_2_ concentration, a possible leakage problem would result in an ingress of O_2_ in the package, causing an increase in microbial proliferation. In such a case, food quality can be monitored through the use of O_2_ and CO_2_ sensors. The leakage indicator usually consists of two components: the color component detecting the O_2_ leakage by changing its color and the scavenging component which absorbs the excess O_2_ [130]. A typical O_2_ indicator contains a redox-dye (i.e., methylene blue), a base (i.e., NaOH, KOH) and a reducing compound (i.e., a reducing sugar). The oxygen indicator is pink in color in the absence (0–1%) of oxygen and turns to blue in the presence (1–5%) of oxygen. Carbon dioxide indicators are used to detect small amounts of CO_2_. They are based on a fast color reaction indicating the presence of a particular amount of CO_2_ contained within a package.

#### 4.4.4. Use of Intelligent Packaging in Seafood Preservation

Kuuliala et al. [44] developed an intelligent packaging system through the determination of volatile compounds (VC) indicative of spoilage of raw cod stored under two different MA atmospheres and air at 4 and 8 °C. Selected-ion flow-tube mass spectrometry (SIFT-MS) was used for the determination of selected VC and 16S rRNA gene amplicon sequencing was used to identify the cod microbiota. The results suggested that *Photobacterium* spp. was the main microbial species contributing to cod spoilage and VC production. The exponential increase in VC concentration and organoleptic rejection occurred at high TVC values (7–7.5 log cfu/g), leading to the conclusion that monitoring of early spoilage is of paramount importance requiring high sensitivity for low VC concentrations. Sadeghi et al. [131] developed a novel amperometric biosensor for xanthine, based on the entrapment of xanthine oxidase (XOD) onto a nanocomposite film via glutaraldehyde. The developed biosensor was used for the determination of xanthine through the amperometric detection of H_2_O_2_ reduction. The results showed that the developed biosensor showed a fast response time of 8 s to xanthine and a linearity within the range 0.2 to 36.0 μM, with a limit of detection of 0.1 μM. The biosensor was applied for measurement of fish and chicken meat freshness, correlating well with standard microbiological and chemical methods. Devi et al. [132] immobilized XOD through covalent bonding onto boronic acid, electrodeposited on a pencil graphite (PG) electrode, via boro ester linkages. The biosensor showed a very rapid response within 3 s at pH 7.2 and 30 °C and linearity in the range, 0.05 μM to 150 μM for hypoxanthine with a limit of detection of 0.05 μM. It was then successfully used to determine hypoxanthine in fish and other foods of animal origin. Dervisevic et al. [133] developed a novel nanocomposite host matrix based on glycidyl-methacrylate-co-vinylferrocene (GMA-co-VFc) for the enzyme immobilization of XOD. The prepared enzyme electrodes exhibited an optimum response at pH 7.0 and 45 °C after ~4 s with a sensitivity of 16 mAM^−1^. A linear response range (2–86 μM), and a low limit of detection 0.12 μM of the xanthine biosensor gave accurate results in measuring xanthine concentration in the fish meat. Dervisevic et al. [134] developed a sensitive amperometric biosensor by preparing a nanocomposite film based on (GMA-co-VFc) and by the covalent bonding of XOD on the surface of the nanocomposite film. Sensor optimal operational conditions were studied. Linearity range was 2–36 μM with a sensitivity of 0.17 μA/M, response time of ~3 s, and detection limit of 0.17 μM. The resulting xanthine biosensor was successfully applied for the measurement of xanthine content in 5, 8, 10, 13, 15, and 20 day-old fish samples. Kuswandi et al. [135] designed a curcumin-based sensor for the evaluation of TVB-N. The operational principle of the prepared membrane was based on a pH increase, as TVB-N was formed gradually in the package headspace, resulting in a change in color of the sensor initially from yellow to orange, then to reddish orange, indicating product spoilage. Furthermore, it was demonstrated that the membrane response correlated well with bacterial growth in shrimp samples. Finally, the curcumin-based sensor was successfully used in the form of a sticker for real-time monitoring of shrimp spoilage under ambient and refrigeration conditions. Cierpizewski et al. [136] developed and evaluated a freshness indicator for packaged fish. The prepared indicator was based on methyl red (MR) and bromothymol blue (BB), pH sensitive dyes. Its response correlated well with sensory properties and microbial populations of fish. Kuswandi et al. [137] developed a chemical sensor using a treated with HCl polyaniline (PANI) film with a dark green color. As TVB-N forms during fish spoilage the sensor color turns to black. Color changes of the sensor correlated well with TVB-N levels of milkfish samples. The PANI film response also correlated well with TVC and *Pseudomonas* spp. growth patterns. Chun et al. [138] developed a freshness sensor consisting of a polymeric matrix solution containing the pH-sensitive dye, bromocresol green, to monitor color changes due to the production of volatile compounds (TMA) during fish spoilage. Mackerel fillets were inoculated with *P. fragi* and stored at different refrigeration and abuse temperatures. Gradual color changes of the freshness sensor response correlated well with populations of *P. fragi*, pH changes and the quality of fish during storage. Giannoglou et al. [139] modeled the kinetics of frozen blueshark and squid quality deterioration and selected an appropriate TTI to monitor them in the frozen chain. Two different (photochromic and enzymic) UV activatable photochromic TTIs were studied and modeled based on the intensity of UV-activation or enzyme concentration. The TTI response was tailored to the expected product shelf life. The selected TTI predicted product shelf life along the cold chain simulating actual temperatures involved. Finally, the effectiveness of TTI for monitoring specific frozen seafood products was validated. Tsironi et al. [140] applied predictive models for *V. parahaemolyticus* (*Vp*) and *V. vulnificus* (*Vv*) growth in oysters. This information was used to design enzymic TTI labels providing a response appropriate to indicate the growth pattern of Vibrio spp. during the transport of oysters from the site of harvest to storage. The results indicated that the *Vv*- and *Vp*-TTIs developed may be cost-effective tools for monitoring the transport of oysters. Tsironi et al. [141] evaluated and validated a TTI for monitoring the shelf life of frozen seafood in the commercial cold chain, covering the time between production and consumption. The study was carried out using frozen blueshark and squid. The attached UV activatable and enzymic TTI was tailored for monitoring quality deterioration of the selected seafood products in the cold chain. The quality and shelf life were estimated through the TTI response. Comparison of predicted values to experimentally measured quality parameter values confirmed the applicability of TTI as an effective indicator of frozen seafood quality during product commercial life.

Based on the current TTI technology the next step forward is the application of TTIs for the management of food safety risks [142]. The USFDA published guidelines in regard to the safe handling of fishery products, including the use of TTIs, to avoid the risk of *C. botulinum* growth and toxin formation [143]. Companies such as Vitsab A.B. (Limhamn, Sweden) and Timestrip UK Ltd. (Cambridge, UK) have designed TTIs which predict the time/temperature conditions required by *C. botulinum* to produce its toxin [144]. Thus, it can be stated that continuous monitoring of temperature using suitable TTIs can reliably predict the safety and quality status of foods, allowing optimization of seafood shelf life within the supply chain. Despite the many benefits of TTIs, the adoption of TTI technology by consumers is very limited. According to Pennanen et al. [145] the basic problem for this is the lack of sufficient knowledge with regard to consumer perception of TTIs. Ongoing studies on the subject show that consumers in numerous countries are gradually considering TTIs to be useful for at least fresh fish, meat, and poultry products, thus, enhancing food safety and security.

### 4.5. Retort Pouch Processing

Retortable flexible containers, in pouch form, were developed in the 1960s through the cooperation of the US Army Natick R&D Command, Reynolds Metals Company and Continental Flexible Packaging. These are flexible, multilayer structures that can be sterilized just like a metal can or glass bottle. Products thermally treated in such pouches are shelf stable for more than one year, without the need for refrigeration. Of the various multilayer combinations developed for this purpose, the most common consists of a three-layer laminate, made inside and out from polyethylene terephthalate (PET)/aluminum foil (Alu F)/cast polypropylene (CPP). Pouches made of PET/Alu F/polyamide (PA)/CPP are also available (Figure 5).

The PET layer contributes with its mechanical strength and printability.The Alu F protects the product from the effect of light and transport of gases, moisture and odors.The PA layer protects from abrasion.The CPP provides heat sealability and acts as a food contact surface.

The above characteristics provide retort pouches (RP) with excellent mechanical and heat transfer properties, a high gas and moisture barrier and high quality sealing properties [147]. The materials used to manufacture RP are FDA approved to undergo sterilization without compromising their physico-chemical and mechanical properties. Commercial RP are available in the form of stand up pouches, spout pouches and zip-lock pouches. The food to be processed is first sealed into the RP, followed by sterilization at 116–121 °C for a predetermined amount of time under pressure inside a retort. The process is analogous to canning or in-bottle sterilization with the tin or glass bottle can being replaced by a less expensive flexible pouch. The RP provides a longer shelf life compared to frozen foods and does not require refrigeration during subsequent storage and distribution. Advantages of the RP over the tin can/glass bottle include [148]:The patented construction of the pouch enables high heat transfer rates for sterilization, resulting in a substantially lower processing time and respective energy consumption.Retention of product nutrient and sensory attributes due to the reduced heat exposure the product undergoes.Reduced preparation time for serving the product (immersion of the pouch in boiling water for 3–5 min. or microwave oven heating).Comparable shelf life of RP products to those in metal containers.No need for refrigeration or freezing by processors, retailers, or consumers.Minimum product–container interaction, without the risk of external corrosion.Easy opening of the pouchReduction in storage space for empty RP for processors. Empty retort pouches occupy 85% less space compared to empty tin cans and are significantly lighter.Less energy required to manufacture pouches compared to metal cans.

#### 4.5.1. Use of RP Packaging in Fish Preservation

Byun et al. [149] used four different RP structures to package shelf stable salmon: (i) CPP, (ii) PET/silicon oxide-coated PA/CPP (SiOx), (iii) aluminum oxide-coated PET/PA/CPP (AlOx) and (iv) PET/Alu F/CPP (FOIL). More specifically, the effect of barrier materials based on thin metal oxide coatings on the product quality was investigated. Of the parameters determined, TBARS was found to be higher in salmon packaged in SiOx RP compared to that packaged in FOIL RP after 8 weeks of storage. Likewise, sensory evaluation showed that salmon packaged in SiO_X_ RP recorded lower scores compared to salmon packaged in FOIL RP after the same period. On the other hand, salmon packaged in FOIL and AlO_X_ RP received similar TBARS and sensory scores. Bindu et al. [150] packaged ‘Fish peera’, an anchovy based traditional product, in a three-layer configuration RP made of PET/Alu F/CPP. The RP was processed in an over pressure autoclave to an Fo value of seven and a cooking time of 66.02 min. Based on biochemical, microbiological, and sensory analysis, the product achieved a shelf life of 1 year during storage at 28 °C. Majumdar et al. [151] packaged Rohu fish in a four-layer laminated retort pouch made of polyester (outer layer)/aluminum foil/nylon/cast polypropylene (inner layer) and sterilized it at 121.1 °C to different Fo values for 7, 8 and 9 min. Sensory, microbiological and texture analysis showed that an Fo value of 8 min with a total process time of 41.7 min at 121.1 °C was the most satisfactory treatment for product preparation. Finally, Majumdar et al. [152] packaged boneless fish balls from Rohu fish in RPs that were the same as described above and processed them in an over-pressure retort at 121.1 °C at three different Fo values of 6, 7 and 9 min. Sensory evaluation, color and texture profile analysis as well as commercial sterility data showed that an Fo = 7 min and a total process time of 42.2 min at 121.1 °C resulted in a product of optimal quality.

#### 4.5.2. Use of RP Packaging in Fishery Product Preservation

Sreelakshmi et al. [153] packaged a ready-to-eat sandwich spread made from crab meat in laminated flexible four-ply pouches consisting of PET (outer ply)/Alu Foil/PA and CPP (inner ply). The product was sterilized at temperatures 111.1, 116.1 and 121.1 °C, to F_o_ values 5, 6, and 7 min. The results showed that the sample processed at 116.1 °C for 6 min with a cook value of 84.29 and a total process time of 42.59 min. received the highest sensory scores.

#### 4.5.3. Edible Films and Coatings/Biodegradable Polymers

According to Dehghani et al. [154] any type of thin edible material used to coat or wrap of a food with the objective of extending product shelf life that can also be consumed along with the food is considered to be an edible film or coating (EFC). As consumers prefer natural food ingredients over synthetic ones, EFCs have become of special interest to both the food research and the food industry. Edible coatings/films prevent moisture and aroma loss from the foodstuff, while simultaneously allowing for permeation of gases, such as O_2_, CO_2_, and C_2_H_4_, which are involved in food product respiration [155]. The main difference between edible films and coatings is that the former are first prepared and then used to wrap the food, whereas the latter are formed directly onto the food surface [156]. By incorporating antibacterial and/or antioxidant agents, edible films and coatings function by delaying microbial spoilage and retarding lipid oxidation, despite the fact that their permeation and mechanical properties are generally inferior to synthetic films. Often, edible films and coatings are used in combination with primary edible packaging or non-edible conventional packaging for proper handling and hygienic purposes [156]. Today, EFCs are used as an inexpensive way to retain the quality and safety of foods, providing a total annual revenue over USD 100 million [155]. EFCs are produced using raw materials such as polysaccharides, proteins and lipids. Film forming and dispersibility/dissolution of raw material in a food grade solvent, preferably water is a basic requirement. The solvent used must also be compatible with potential additives (antioxidants, antimicrobial agents, etc.) that may be incorporated in the edible film/coating.

Polysaccharide-based EFCs can be prepared using a number of natural substrates such as starch, cellulose, pectin derivatives, alginates, carrageenan and agar [157], gums [158] and chitosan [159,160,161]. Polysaccharide coatings and films are generally very hydrophilic and, thus, do not provide protection against moisture but they do have selective permeability to oxygen and carbon dioxide and are resistant to lipid migration. Chitosan and various gums are widely used for the preparation of EFCs. When such biopolymers solubilize they form specific structures known as micelles, stabilized by intermolecular hydrogen bonds. Micelles aid film formation due to their stability during drying.

Using proteins as the raw material, films and coatings have been prepared mainly from cheese whey [162,163,164], corn zein [165,166], collagen, wheat gluten [167] and gelatin [11,168]. Protein film formation is based on the mechanism of denaturation, which is triggered by heat, pH changes or solvents. Denatured polypeptides associate through the formation of new intermolecular interactions [156]. Protein-based films exhibit strong adhesion to the hydrophilic surfaces of seafood, usually providing barriers to gases but not to moisture [169]. In general, water vapor barrier of protein films is approximately 2 to 4× greater than those of synthetic packaging materials.

Lipids, on the other hand, are not macromolecules and do not form cohesive films. They are, thus, used either as coating materials or added directly to natural polymers to form composite films, which due their non-polar nature they improve the film moisture barrier [156]. Incorporation of lipid-based substances into EFCs also improves film cohesiveness and flexibility. This can aid in extending the freshness, aroma, color, texture, and microbiological stability of fresh and processed seafood [170,171,172]. Within lipids, waxes are esters of long-chain carboxylic acids with long-chain aliphatic alcohols. Waxes are non-polar substances thus, enhancing moisture retention in seafood products [139,156]. Mono-, di- and long chain (non-polar) triglycerides are also used as coating materials providing a barrier to water vapor permeability. Furthermore, the addition of different active compounds into the coatings such as organic acids and their salts, (i.e., benzoic, propionic, lactic, sorbic and acetic, sodium lactate, sodium diacetate and potassium sorbate), EOs/plant extracts (oregano, thyme, cumin, cinnamon, lemongrass), bacteriocins (nisin), proteins and/or chitosan, may substantially affect lipid oxidation, and autolytic and microbial deterioration [169].

#### 4.5.4. Use of EFCs/Biodegradable Polymers in Seafood Preservation

Chitosan-based antimicrobial films have been repeatedly applied with considerable success as EFCs to preserve a variety of fish and fishery products: Nowzari et al. [173] investigated the effect of chitosan-gelatin (Ch-G) coating and film on the development of rancid off-odors in refrigerated, filleted trout stored for 16 days. Microbiological and biochemical quality parameters of both coated and film wrapped fish samples were determined over storage time. Results showed that Ch-G coating and film wrapping maintained fish quality and resulted in product shelf life extension. The coating was more effective compared to the film for the reduction of lipid oxidation of fillets. Negligible differences were recorded between the two treatments with regard to the control of bacterial contamination. Soares et al. [174] evaluated the preservation effect of coating of frozen salmon with chitosan solutions compared to that of water glazing. Salmon was stored at −5 °C for 14 weeks. It was shown that chitosan coatings resulted in the protection of frozen fish from spoilage. TVC and TVB-N remained below 5 × 10^5^ cfu/g and 35 mg N_2_/100 g fish (recommended max. limits), respectively [175,176]. The higher the concentration of chitosan solution used, the greater was the preservation effect. Günlü and Koyun [177] investigated changes in rainbow trout fillet quality parameters stored at 4 °C as a function of chitosan-based EFC, VP and HHP applied. Sample treatment included: VP (serving as control, C), HHP processing followed by VP (HHP), VP after wrapping in chitosan-based film (CFW) and HHP processing after VP and wrapping in chitosan-based film (HHP+CFW). Results showed a 4 day extension in product shelf life in HHP group, 8 day extension in CFW group and 24 day extension in HHP+CFW group in comparison to the control group. Khemir et al. [178] applied a chitosan-microparticles (CM)-coating (CMC) to sea bream fillets at concentrations 0.2 and 0.5% (*w*/*w*) which were then vacuum packaged and kept under refrigeration for 24 days in order to determine product shelf life. TVB-N, TMA and TBA recorded higher values in control in comparison to CMC treated samples, exhibiting a concentration-dependent effect. The shelf life of the 0.5%-CM treatment was 12 days, versus 6 days for the controls.

Various organic acids and their salts [e.g., citric acid, sodium lactate (SL), sodium diacetate (SDA), and potassium sorbate (PS)] as well as plant extracts (e.g., grape seed extract, tea polyphenols and licorice extract) have been used to prepare chitosan based films with antimicrobial properties. Jiang et al. [179] investigated the efficacy of chitosan-based EFCs containing (SL), (SDA) and (PS) for the inhibition of *L. monocytogenes* growth on cold-smoked refrigerated salmon. Results showed that the combined treatments (chitosan+organic acids) and (chitosan+plant extracts) were more effective than chitosan films alone for the reduction of *L. monocytogenes*. Li et al. [180] studied the influence of grape seed extract (GSE) and tea polyphenols (TP), in combination with chitosan (Ch), on shelf life of red drum fillets stored under refrigeration. Results showed that the two treatments (Ch+GSE and Ch+TP) effectively maintained quality and extended product shelf life by 6–8 days in comparison to control samples. Qiu et al. [181] evaluated the preservation effect of chitosan (Ch), chitosan plus citric acid (Ch+CA), chitosan plus licorice extract (Ch+LE) on refrigerated sea bass fillets for a period of 12 days. Results showed that (CA) or (LE) significantly retarded lipid oxidation and inhibited microbial growth as shown by TBARS and TVC values compared to (Ch) alone. Sensory scores and TVB-N values indicated that the combinations (chitosan+citric acid) and (chitosan+ licorice extract) significantly extended product shelf life. Souza et al. [182] coated salmon fillets with chitosan with the objective to extend product shelf life. Coated fillets were stored at 0 °C for 18 days. Results indicated that fish samples coated with chitosan maintained lower pH and K values on day 6 of storage and lower TVB-N, TMA-N, and MDA values on day 9 of storage in comparison to control samples. A slower increase in TVC was also observed in the coated fish. The study concluded that chitosan-based coatings extended the shelf life of salmon fillets by 3 days in comparison to controls. Duan et al. [183] coated lingcod fillets with chitosan containing fish oil rich in Eicosapentaenoic acid (EPA) and Docosahexaenoic acid (DHA). Fillets were stored at either 2 °C for 3-weeks or at −20 °C for 3-months. The study showed that the chitosan coatings alone reduced TVC and psychrotrophic counts in both sample treatments. Chitosan–fish oil coating increased omega-3 FA content of fish, decreased lipid oxidation in sample treatments, and also decreased drip loss in frozen samples. Choulitoudi et al. [184] investigated the antioxidant and antimicrobial activity of rosemary EO (REO) and extracts, immobilized on carboxyl methyl cellulose (CMC) edible coating for smoked eel. Analysis of the (REO) identified 1,8-cineole, l-camphor, a-pinene, and 1-borneol, as the main constituents. Adding the extract in CMC coating exhibited antioxidant protection to smoked eel, which was concentration dependent. Furthermore, the combination of the extract +REO significantly reduced primary and secondary oxidation products. The highest antimicrobial activity was obtained by using both REO and extracts at a concentration of 800 ppm concentration resulting in the decrease of TVC, Pseudomonas spp., and LAB growth.

The film forming properties of the specific proteins of animal and plant origin and their effect on the preservation of fish and fishery products have recently been studied. Jiang et al. [185] prepared an antimicrobial coating made of catfish skin gelatin with the incorporation of potassium sorbate, sodium tripolyphosphate, or both ingredients. The coating was applied to fresh white shrimp stored in ice aerobically. The study showed that the antimicrobial coating suppressed microbial proliferation extending shrimp shelf life to 10 days. Feng et al. [186] reported that chitosan/gelatin coatings significantly reduced spoilage of golden pomfret fillets stored under refrigeration. Using Matrix-Assisted Laser Desorption Ionization-Time of Flight Mass Spectroscopy (MALDI-TOF MS) they demonstrated that the coatings retarded protein decomposition of fillets. Dursun and Erkan [167] studied the effect of protein based edible coating in combination with VP on the shelf life extension of hot-smoked, refrigerated rainbow trout stored for 6 weeks. Soy protein isolate (SPI), whey protein isolate (WPI), egg white powder protein (EP), wheat gluten (WG), corn protein (Z), gelatin (G), collagen (Co) and from protein concentrate of two different fish species [rainbow trout (RT) and Atlantic mackerel (AM) were used to prepare edible coatings. Organoleptic, microbiological, biochemical, color and texture parameters were monitored during storage. Of all the above coatings, treatment (Co) succeeded in extending product shelf life by 2–3 weeks. Kim et al. [187] carried out a study for the optimization of processing conditions of coating in order to prevent oxidation in boiled-dried anchovy using response surface methodology (RSM). The application of RSM showed that the optimum ultra-filtrates of the rockfish skin gelatin hydrolysate (FGH) and pre-drying time for boiled anchovy were 4.6% and 180 min, respectively. PV and TBA values of the coated boiled-dried anchovy after drying, were lower than those of the uncoated anchovy. Lin et al. [165] employed three synthetic antioxidants (BHA, BHT and PG) to formulate an antioxidant coating based on zein for fish balls (a surimi product, high in lipid and protein content). Quality parameters of refrigerated fish balls monitored, included PV, TBARS and weight loss. Of the three types of antioxidant containing coatings, the (PG) containing zein coating was more effective in quality retention compared to its BHA- and BHT-containing counterparts. Rodriguez-Turienzo et al. [163] added transglutaminase (TGa) to heated (H) or ultrasound-treated (UT) whey protein coatings and studied their effect on the quality of frozen Atlantic salmon. The addition of TGa did not affect the yields, drip loss, color or chemical composition of the fish fillets. Yields were higher in (H) treated coatings compared to those of the (UT) treatment. TGa addition to (H) whey protein coatings retarded fat oxidation. (UT) coatings with or without the enzyme, had a similar effect on the reduction of product fat oxidation.

Peptides from LAB that inhibit the growth or even kill specific groups of Gram positive microorganisms are known as bacteriocins. *Lactococcus lactis* spp. produce nisin, a natural antimicrobial agent, listed as GRAS by the USFDA. Among bacteriocins, nisin is presently the only one approved to be used as a food additive in numerous countries. It is active against Gram(+) bacteria, and when combined with a chelating agent, such as ethylenediaminetetraacetic acid (EDTA), it also exhibits considerable activity against Gram(−) bacteria [188]. Lin et al. [166] used edible zein coatings with the addition of nisin or nisin/EDTA to extend the shelf life of commercially prepared fish balls. The results showed that fish balls with the antimicrobial coating, recorded a TVC <1 log cfu/g, after 15 days under refrigeration, whereas uncoated samples reached a TVC value of about 3 log cfu/g on the same day. Likewise, coated fish balls recorded significantly lower TVB-N values in comparison to the control. Coated fish balls, with or without antimicrobial agent, exhibited significantly lower weight loss than their uncoated counter parts.

Lactoperoxidase (LPO) is an enzyme secreted by the mammary, salivary and lacrimal glands of mammals. It is composed of a single polypeptide chain and exhibits a broad spectrum of antimicrobial activity. LPO has a bactericidal effect on Gram(−) bacteria and a bacteriostatic effect on Gram(+) bacteria, in addition to antifungal and antiviral activities [189]. Jasour et al. [190] incorporated LPO into a chitosan coating solution (CH) and evaluated the coating effect on the shelf life of refrigerated rainbow trout stored for 16 days. The study showed that samples treated with CH+LPO had lower counts of *S. putrefaciens*, *P. fluorescens*, and psychrotrophic and mesophilic bacteria compared to the CH and control group treatment throughout storage. TVB-N values for the CH+LPO samples remained below the proposed limit of (30–35 mg N/100 g), while the CH and control groups reached this value on days 12 and 16, respectively. The coating treatments (CH and CH+LPO) extended the product shelf life by 4+ days compared to controls. Shokri et al. [189] incorporated LPO at various concentrations in a whey protein solution and tested its preservation potential on refrigerated rainbow trout fillets during storage for 16 days. The control and 1.25% LPO-treatment developed high rancidity on day 12 and 16, respectively, whereas the higher LPO concentrations gave better sensory characteristics, lower chemical index (TVB-N and pH) values and lower microbiological parameters (TVC, psychrotrophic count, *Pseudomonas* spp. and SSO) values. The LPO treatment did not affect lipid oxidation in the fillets. Shelf life was extended by at least 4 days with the addition of the LPO. Shokri and Ehsani [191] evaluated the effect of a whey protein coating solution (WPS) containing 2.5% LPO and α-tocopherol (1.5% and 3%) on maintaining the quality of pike-perch fillets stored under refrigeration. Treatment with α-tocopherol exhibited better color, odor, and lower TBA values in comparison to WPS treatments. WPS+2.5% LPO treatment showed lower TVC and TVB-N values, as well as better texture and overall acceptability scores in comparison to WPS, with no significant changes recorded in color, odor, and TBA in the LPO treatment. WPS+2.5% LPO+1.5% α-tocopherol and WPS+2.5% LPOS+3% α-tocopherol treatments resulted in a higher quality product compared to other treatments. Farshidi et al. [192] coated shrimp samples by immersion into a whey protein solution containing various concentrations of LPO and evaluated the preservation potential of the system. Coated or uncoated shrimp were stored under refrigeration for 16 days. The results showed a decrease in spoilage specific organisms and TVB-N values with the addition of LPO. TBA was not affected by the addition of LPO. The levels of LPO correlated well with sensory evaluation results. Barkhori et al. [193] studied the effect of a functional alginate coating containing LPO and *Zataria multiflora* essential oil (ZEO), individually and in combination with natural additives, on the quality characteristics of refrigerated rainbow trout fillets for 16 days. The results showed that the combined use of ZEO and LPO had the strongest effect on chemical spoilage parameters (TVB-N, pH) and spoilage organisms of trout fillets during storage; however, the addition of ZEO or LPO individually to samples produced statistically significant effects, enhancing quality of trout fillets. Cai et al. [160] investigated the combined effect of a chitosan and ergothioneine coating (CH+ER) on the keeping quality of refrigerated sea bass stored for 16 days. Treatments included: control without coating, (CH) with chitosan coating, (ER) with ergothioneine immersion, and (CH+ER) with chitosan containing ergothioneine coating. The results indicated that treatment with (CH+ER) coating decreased TVB-N, PV, and TBA values and microbial counts, including the pseudomonads, compared to controls. Furthermore, the biogenic amine content, especially putrescine, cadaverine, and histamine decreased in the sea bass treated with the chitosan-ergothioneine coating. In agreement with the microbiological and chemical analyses, sensory analysis confirmed the efficacy of the chitosan-ergothioneine coating to maintain the quality characteristics of sea bass during storage. In a similar experiment, Cai et al. [158] evaluated the effect of gum Arabic coating (GA) with and without the addition of ergothioneine (GA+ER), on the organoleptic quality and physicochemical parameter values of red sea bream stored under refrigeration for 16 days. The results showed that the combined treatment (GA+ER) suppressed protein degradation, nucleotide breakdown, fat oxidation, and protein decomposition, while reducing microbial growth in comparison to the controls. The positive effect of (GA+ER) was stronger than that of individual GA or ER treatment. Sensory analysis confirmed the efficacy of the (GA+ER) coating to maintain the overall quality of red sea bream during refrigerated storage.

Cinnamon and nisin added to an alginate-calcium coating were used by Lu et al. [194] for the purpose of maintaining the keeping quality of refrigerated snakehead. Control (CK) snakehead fish fillets, fillets treated with the alginate-calcium coating (YO), alginate-calcium coating containing cinnamon (Y1), alginate-calcium coating containing nisin and EDTA (Y2), or alginate-calcium coating containing cinnamon, nisin and EDTA (Y3) were evaluated. Treatments Y1 and Y3 were more effective for the inhibition of bacterial growth and retention of physico-chemical parameter (pH, TVB-N and TBA) values of snakehead fish compared to CK, YO and Y2. The study concluded that treatment Y1 was efficient in maintaining product quality during storage; however, the color of fish fillets of Y1 and Y3 were unavoidably changed due to the color of cinnamon. In a similar study, Andevari and Rezaei [195] used cinnamon EO (CEO) in gelatin coatings to preserve the quality of refrigerated rainbow trout fillets for a period of 20 days. The results showed that the application of gelatin coating with cinnamon to trout fillets, reduced the TVC after 15 days of storage. Likewise, fish fillets with gelatin coating containing cinnamon EO resulted in lower TVB-N values than gelatin-coated fillets and controls up to day 15 of storage. Aşik and Candoǧan [196] evaluated the effect of chitosan-based edible coatings (CC) containing garlic oil (GO) at three different concentrations on shrimp quality under refrigeration for a period of 11 days. CC led to a reduction in TVC up to 2 log units. Antioxidant activity of CC was observed only during the early stages of storage, while, interestingly, GO enhanced lipid oxidation. At a concentration of 0.5% GO into the chitosan coating, the CC+GO treatment extended shrimp shelf life under refrigeration. Heydari et al. [197] evaluated the effect of a sodium alginate (SA) coating containing horsemint essential oil (HEO) on the keeping quality of bighead carp fillets stored under refrigeration. The SA coating enriched with (HEO) delayed fillet spoilage and extended the product shelf life. SA-HEO treated samples resulted in lower TVB-N, PV, TBA and FFA values during refrigerated storage in comparison to the SA and controls. This combination treatment also led to a reduction in TVC of fillets by ca. 1.5 log cfu/g compared to (SA). Karami et al. [198] incorporated *Ziziphora clinopodioides* EO (ZEO) and sesame oil (SO) into a chitosan-flaxseed mucilage (CH-FM) edible film and evaluated its effectiveness against *L. monocytogenes*, *S. typhimurium*, *St. aureus* and *E. coli O157:H7* in minced trout fillets stored under refrigeration. The main constituents of ZEO included carvacrol, thymol, ɣ-terpinene and p-cymene. The weakest antimicrobial effect against *St. aureus*, *L. monocytogenes*, *E. coli O157:H7* and *S. typhimurium* was observed for CH-FM films containing 0.5% SO, while the strongest was observed for CH-FM films containing 0.5% ZEO + 0.75%. SO. The antioxidant activity of CH-FM based films ranged from 5.5% to 37%. After two weeks of storage, the counts of *L. monocytogenes*, *St. aureus*, *E. coli O157:H7* and *S. typhimurium* were significantly lower in treated trout fillets compared to control groups. Kim et al. [199] prepared an antioxidant film based on defatted mustard meal (DMM) and evaluated its coating potential for smoked salmon. With the addition of 5% xanthan, the composite film exhibited antioxidant properties retarding lipid oxidation when stored at 4, 10, and 20 °C for 21 days. Panelists preferred the coated to uncoated salmon in terms of glossiness and fish odor.

Shrimp processing is usually done manually with the risk of product contamination by pathogenic microorganisms, especially following thawing. Guo et al. [200] investigated the effect of antimicrobial-coating treatments in combination with cryogenic freezing for growth inhibition of *Listeria innocua*, a substitute for *L. monocytogenes*, on ready-to-eat (RTE) shrimp. *L. innocua* was inoculated at a population of 3 log cfu/g onto cooked RTE shrimp, subsequently coated with chitosan containing allyl isothiocyanate (AIT), or lauric arginate ester (LAE). Treated shrimp were then kept at −18 °C for 6 d before thawing. Results showed that *L. innocua* population decreased by 1–5.5 log cfu/g by application of the antimicrobial coatings. No synergistic effects between coating and cryogenic freezing were observed against *L. innocua*. García-Soto et al. [201] packaged megrim in a polylactic acid (PLA) film containing lyophilized alga *Fucus spiralis* and sorbic acid and investigated its effect on product keeping quality during refrigerated storage. Sensory evaluation showed that PLA film wrapped samples containing 0.8% alga and 1% sorbic acid maintained acceptable quality up to day 11, while controls (fish wrapped in polyethylene film) were rejected on the same day. A preservative effect of the specific treatment was also shown by the determination of chemical indices: TMA-N and PV. Javidi et al. [202] evaluated the mechanical and antimicrobial properties of PLA biodegradable films containing oregano essential oil (OEO). The PLA-EO films had a higher flexibility than straight PLA films. The antimicrobial properties of the film were improved by incorporating EO. The efficacy of the active film containing 1.5% (*w*/*w*) EO was tested to extend the shelf life of rainbow trout during chilled storage. The results showed that the PLA +OEO film retarded the growth of both spoilage microorganisms and pathogens, i.e., *St. aureus* and *E. coli*. The study suggests that the PLA films developed may be used to design antimicrobial packaging materials. Cardoso et al. [203] prepared biodegradable antimicrobial films made of poly(butylene adipate-co-terephthalate) (PBAT) with the addition of oregano essential oil (OEO) applied to fish fillets during storage. Films were hot-melt extruded and their antioxidant and antimicrobial activities were determined. The results showed that a higher OEO content resulted in higher water vapor permeability. Microbiological analysis demonstrated the efficacy of the films to reduce total coliforms, *St. aureus* and psychrotrophic microorganisms. With regard to film antioxidant properties, it was shown that as the OEO concentration increased, so did the antioxidant activity. The authors concluded that prepared films proved to be an efficient active packaging system for the control microbial growth and retardation of lipid oxidation in fish fillets. Gómez-Estaca et al. [168] evaluated the effect of biodegradable packaging on the quality of cod fillets. Samples were packaged in chitosan-gelatin films (Ch-ge) containing clove EO (CEO) and stored under refrigeration. The treatment resulted in a drastic reduction in enterobacteria as well as other Gram(−) bacteria. In contrast, LAB populations were not affected during most of the storage period. Biochemical indices values were in close agreement with the microbiological data, leading to the suggestion of the potential use of these films for preserving fish. Kakaei and Shahbazi [204] used bio-composite Ch-ge films containing 1% grape seed extract (GSE) and/or 2% *Ziziphora clinopodioides* EO (ZEO) and assessed its potential for extending the shelf life of minced trout fillets stored under refrigeration for 11 days. Samples packaged in Ch-ge films containing different amounts of GSE and/or ZEO exhibited delayed spoilage compared to the control group. Treatments ZEO-2%+GSE-1% and ZEO-2%+GSE-2% gave the highest organoleptic acceptability values. The latter combination resulted in the lowest bacterial growth, TVB-N and PV values. Merlo et al. [205] packaged filleted skinless salmon in Ch films containing pink pepper extracts (CFPP) under MAP (100% CO_2_) and evaluated product quality during refrigerated storage for a period of 4 weeks. The same treatment excluding the pink pepper extract was also evaluated (CF). Both treatments significantly retarded fat oxidation compared to the control sample. TVC was significantly lower in CFPP, resulting in lower TMA values. CFPP also showed the highest sensory score. Compared to CF, CFPP resulted in higher quality and longer shelf life of salmon fillets. Mohan et al. [206] applied an edible chitosan (CS) coating to preserve ice-stored Indian oil sardines. Uncoated controls exhibited a 5 day shelf life, while the CS coated fish exhibited a shelf life of 8 and 10 days for 1% and 2% CS coated fish, respectively. CS coated samples also exhibited a lower TVC, fat oxidation, TMA and TVB-N compared to controls. In a similar experiment Fan et al. [207] coated frozen silver carp with 2% chitosan in an effort to retard product spoilage. The results of chemical, microbiological and sensory analysis showed the effectiveness of chitosan to maintain acceptable quality of frozen carp for over four weeks. Alak [208] prepared chitosan coatings for brown trout by dissolving chitosan in acetic acid (AC) and lactic acid (LA). The treatment with AA proved more effective for the preservation of trout resulting in lower TVC, pseudomonads, LAB, pH, TVB-N, and TBARS values in comparison to LA. Carrión-Granda et al. [209] coated peeled RTE shrimp tails with Ch containing 0.5% oregano and thyme EOs (OEO, TEO). Products were MA packaged and kept under refrigeration for 12 days. Ch containing TEO resulted in a higher level of inhibition of psychrotrophic bacteria and LAB in shrimp than OEO. Yanar et al. [210] tested commercial Ch or chitosan produced from *Metapenaeus stebbingi* shells and reported that both treatments significantly reduced the level of FFA, TBA and PV in refrigerated European eel. Mohan et al. [206] evaluated the effect of Ch edible coating (1% and 2%) on the quality characteristics of filleted Indian oil sardines during chilled storage. The Ch coating retarded bacterial growth and significantly reduced TVB-N and oxidation products. A value of 2% Ch resulted in a greater reduction in TVB-N and TMA than 1%. The Ch coating also improved the water holding capacity, drip loss and textural properties of treated samples compared to untreated counterparts. Additionally, 1% chitosan treated sardines had a shelf life of 8 days vs. 10 days for the 2% chitosan treated samples. The shelf life of controls was only 5 days. Vatavali et al. [211] studied the effect of Ch combined with oregano EO dip on the shelf life of whole red porgy stored in ice/refrigerated for 20 days. Microbiological, physicο-chemical and sensory attributes were monitored over time. TVB-N values exceeded 30–35 mg N 100 g^−1^ (proposed acceptability limit) on day 13, 15–16 and 20 for controls, samples treated with OEO and samples treated with Ch. Samples treated with Ch+OEO never reached this value at the end of the 20 day period. TMA values exceeded 5–6 mg N 100 g^−1^ (proposed acceptability limit) on day 11–12 for controls, day 14–15 for OEO treated samples, day 19–20 for CS treated, and >20 days for CS+OEO for treated samples. TBARS values for all treatments remained ≤0.4 mg MDA kg^−1^ at all times during storage. Sensory evaluation data showed a product shelf life of ca. 11 days for controls, 16 days for OEO treated samples, 18 days for Ch treated samples and 19–20 days for Ch+OEO treated samples.

Bazargani-Gilani [212] activated a sodium alginate-based edible coating (AL) with resveratrol (RE) as a dietary supplement to extend the shelf life of rainbow trout fillets stored under refrigeration for 15 days. Sample treatments included: AL, AL–RE 0.1%, AL–RE 0.2%, and control. All treatments showed a marked reduction in TVC, psychrotrophs, *Pseudomonas* spp., LAB, Enterobacteriaceae and yeasts–molds compared to the control during storage. TVB-N and TBA values were also significantly lower in all treatments in comparison to the control. The addition of RE resulted in an increase in product shelf life. The study concluded that RE can be considered as a natural antioxidant/antimicrobial for the preservation of rainbow trout fillets. Cai et al. [213] determined the shelf life and quality changes in red sea bream coated using sodium alginate (SA) containing 6-gingerol (GR) stored under refrigeration for 20 days. The results of the physico-chemical and sensory analysis showed that the SA+GR coating retarded fat oxidation, protein and nucleotide breakdown, and microbial growth in comparison to the control. Sensory evaluation showed that red sea bream maintained its overall quality by application of the SA+GR coating during the entire storage period. Jalali et al. [115] investigated the effect of an alginate/carboxyl methylcellulose composite coating (C-A) containing clove essential oil (CEO) on the quality of refrigerated silver carp fillets stored for 16 days. Treatments included: (C-A); C-A + 1%; CEO C-A + 1.5% CEO and the control. The above treatments were also tested for their efficacy to control the growth of *Eschershia coli O157:H7* inoculated in the fillets. The results showed that the combination C-A + CEO 1.5% resulted in the most acceptable microbiological, biochemical and sensory attributes throughout storage at 4 °C. This treatment also reduced the population of *E. coli O157:H7* < 2 log cfu/g (acceptability limit) for the entire storage period.

Concha-Meyer et al. [214] prepared an antimicrobial alginate film (AL) containing two strains of LAB and nisin for controlling the growth of *L. monocytogenes* in vacuum packaged cold-smoked salmon. Pieces of salmon, inoculated with *L. monocytogenes* at a population of 10^4^ cfu/cm^2^, were wrapped with the AL film + LAB strains + nisin and stored at 4 °C. *L. monocytogenes* colonies were enumerated in weekly intervals up to 4 weeks to determine pathogen inhibition. After 4 weeks, salmon pieces wrapped in the AL film without the LAB strains + nisin showed an increase of 2.4 log cycles in *L. monocytogenes* growth. On the other hand, films with either of the two LAB strain or the combination of nisin + both LAB strains exhibited a bactericidal effect on the pathogen during the entire storage period, which exceeded the commercial standard for the shelf life for smoked salmon. The results clearly showed that experimental films inhibited the growth of *L. monocytogenes* on salmon under refrigeration.

### 4.6. Safety Concerns and Legal Aspects of Packaging

#### 4.6.1. Vacuum Packaging and Modified Atmosphere Packaging

VP or MAP of fishery products stored under refrigeration have been associated with serious food safety related problems. The main safety concerns involve refrigeration conditions enabling outgrowth and toxin production of non-proteolytic types B, E, and F *C. botulinum* which commonly occur in fishery products. It has been repeatedly documented in the literature that botulinum toxin production may occur in seafood before rejection by consumers due to apparent spoilage. The National Advisory Committee on Microbiological Criteria for Foods of the Food Safety and Inspection Service of the USDA [215], having reviewed microbiological safety issues related to VP and MAP of refrigerated raw fishery products, as soon as 1992, adopted primary and secondary preventive measures against food pathogens including mainly *C. botulinum* but also *Y. enterocolitica*, *L monocytogenes* and histamine production. In summary, the committee has recommended that restricted use of VP/MAP should be made for refrigerated raw fishery products only (i) in conjunction with the implementation of a detailed HAACP plan as a safeguard measure, (ii) when using high quality raw fish, (iii) when products are held ≤38 °F/3.3 °C at all stages of preparation and distribution, (iv) when it is certain that sensory rejection by the consumer precedes the possibility of toxin production and (v) in addition to refrigeration, secondary measures are employed to further assure product safety (i.e., appropriate labeling, etc.). Similar legislation has been adopted in Europe through Regulation EC 852/2004 [216] and the Food Standards Agency (London, UK) [217].

#### 4.6.2. Active (AP) and Intelligent Packaging (IP)

According to PIRA International, the combined world value of the AP and IP market in 2005 was 1.8 mllion USD, with oxygen scavengers representing 40% and moisture absorbers accounting for 15% of this amount [218]. To enter the European market, each AP or IP system should comply with European legislation. AP and IP systems are made up of (i) a non-active part and (ii) an active part. The non-active part, i.e., the polymeric packaging material(s) should comply with the food contact legislation included in Directive 2002/72/EC [219]. AP and IP systems involving different materials, i.e., ceramics, cellophane, etc., should comply with Regulations (EC) 1935/2004 [220] and 450/2009 [96]. In summary, Regulation (EC) 1935/2004 [220] states that: (i) constituents of food contact materials shall not be transferred to food in amounts which could: endanger human health or result in unacceptable changes to food composition or cause any deterioration in food sensory attributes, (ii) adequate labeling must be provided to identify any non-edible parts and to indicate that the materials used are truly active and/or intelligent and (iii) information should be provided to food packers and consumers on the appropriate and safe use of active and intelligent materials and articles [221].

Regulation 450/2009 [96] provides additional information included in Regulation 450/2009, summarized as follows: (i) components of active and intelligent materials and articles should only be substances included in the ‘community list’ of authorized substances (ii) substances added or incorporated by techniques such as grafting or immobilization as well as migrating active substances shall comply with the relevant community and national legislation and Regulation (EC) No. 1935/2004 [220], (iii) the amount of a migrating active substance shall not be included in the value of the overall migration (OML), in cases where the latter is established in a specific community measure for a given food contact material in which the active substance is incorporated (iv) the migration into food of the substances from components which are not in direct contact with food or the environment surrounding the food (as referred to in Article 5) shall not exceed 0.01 mg/kg, measured with statistical certainty by a method of analysis in accordance with Article 11 of Regulation (EC) No. 882/2004 [222], (v) non-edible parts, active and intelligent materials and articles or parts thereof shall be labeled, whenever they are perceived as edible:(a)with the words ‘DO NOT EAT’ and(b)always where technically possible, with the following symbol (Figure 6):

With regard to compliance testing, Directives 82/711/EEC [223] or 85/572/EEC [224] set specific requirements for migration testing from food packaging materials in contact with packaged food. A basic difference between active and intelligent systems placed outside or inside the primary package is that systems placed on the outer surface the package need no migration testing as the packaging material will act as a ‘functional barrier’ which is expected to substantially reduce the potential migration. In addition to the above, most intelligent systems attached on the package outer surface refer to foods usually stored for short periods of time at low temperatures. In this case, the risk of exceeding a lag-time in diffusion through the packaging material is negligible. Nevertheless, a judgement and a decision should be made specifically for each case. If, on the other hand, the active or intelligent additive is incorporated into the packaging container, i.e., plastic film, then such a component should comply with the rules included in the respective EU Directives. If the active or intelligent ingredient is inserted in the primary package in the form of a sachet, label, sticker or sheet, the food/contact area ratio is large and the actual contact is limited. For such cases, in order to establish the necessary legal requirements, the physical state of the food in contact with the active/intelligent ingredient should be considered. The packaged food can be (i) dry, i.e., cereals, bakery products, legumes (ii) liquid, i.e., milk, fruit juices beer, or (iii) moist (semi-solid foods), i.e., meat, poultry, fish, soft cheeses. In case of dry foods, specific migration of individual substances should be determined directly in the food, as no tests have been developed involving simulants. Liquid food testing should be carried out as described in Directives 82/711/EEC [223] and 85/572/EEC [224]. Finally, in the case of semi-solid foods, testing protocols should be developed [225].

A typical example is an oxygen scavenger (OS) based on the oxidation of Fe^2+^ to Fe^3+^ used to protect fatty fishery products from lipid oxidation. Based on Directives 82/711/EEC [223] and 85/572/EEC [224] in order to determine either the overall or specific migration, the OS should be immersed in food simulants (i.e., water or 3% acetic acid) and stored for the prescribed time-temperature conditions. This, of course, is not a realistic condition. For such cases, a specific ‘dedicated’ test has been proposed to determine overall migration using filter paper soaked with the simulant (Figure 7). According to this specific test, the food sample is wrapped with a filter paper, soaked with simulant and sandwiched between two glass plates. The possible weight of the food is simulated by a total weight of 70 g (50 g weight plus 20 g weight of the glass plate) placed on top of the OS [221]. The whole package setup is stored in the oven for the prescribed time–temperature conditions. After completion of the storage period, the filter paper is removed and extracted with the food simulant to determine the overall or specific migration. Comparison of results of the above test to those described in Directives 82/711/EEC [223] and 85/572/EEC [224] has shown much lower overall migration values for the former versus the latter.

#### 4.6.3. Edible Films and Coatings

As a basic requirement, edible films and coatings (EFCs), including all components must be safe for consumption or must be listed as GRAS [226]. Furthermore, EFCs, when containing antimicrobials, are considered as active food packaging materials. In this case, they should comply with Regulations (EC) 1935/2004 [220] and 450/2009 [96]. EFCs are designed to function as carriers, preferably for the controlled release of active compounds from the package into the contained food. EFCs can be categorized as (i) food ingredients, (ii) food additives, (iii) food contact substances or (iv) food packaging materials; thus, their use should comply with all regulations referring to food ingredients, since, eventually, they become a part of the edible portion of the food consumed [227]. Furthermore, besides the above requirement of GRAS status of all ingredients used, EFCs should be commercially produced in processing plants following good manufacturing practice (GMP) [228]. In addition to the above, EFCs based on all biodegradable polymers are encouraged to be designed and produced considering ‘green-packaging’ initiatives, such as those of the Environmental Protection Agency (EPA), that is, (i) reducing the amount of packaging materials used at the source and (ii) reusing/recycling/composting used packaging materials [229].

## 5. Conclusions and Future Outlook

Packaging has come a long way since the late 1960s when packaging as a science was introduced into university studies, initially in the USA and later on in Japan, Europe and Australia. The initial concept of food packaging, that is, a passive means to contain and protect the contained food product from the effect of oxygen, moisture, light and biological contamination has given way to more sophisticated functions, such as the interaction of product-package, which: (i) in the form of oxygen and ethylene absorbers, carbon dioxide and ethanol emitters antioxidant and antimicrobial packaging materials in combination with MAP, VP and vacuum skin packaging, have led to the considerable shelf life extension of the contained food product; (ii) in the form of freshness, time-temperature and leakage indicators provide information on product quality as a result of either microbial growth or biochemical changes occurring within the packaged food product. However, with the exception of MAP, VP and retort pouch technologies, which have found commercial acceptance, active and intelligent packaging as well as edible films and coatings are still in the research stage and require further development and testing before they become commercial. At the same time, the developments and usage of biodegradable packaging materials has more recently been contributing to the management of problems related to environmental pollution, which threatens numerous ecosystems.

As seafood represents one of the most-traded commodities of the world food sector it is anticipated that, in years to come, seafood packaging will continue to grow in areas such as the introduction of new bio-based eco-friendly packaging materials with or without natural additives and improved functional properties, new packaging technologies resulting in the maintenance of food product quality and safety with extended product shelf lives, which, upon use, will have a reduced carbon footprint on the environment. Prerequisites for such developments include (i) that the sensory/consumer acceptance properties are not affected and (ii) all raw materials and additives used in specific applications should comply with the already existing relevant legislation or new legislation to be introduced.

## Figures and Tables

**Figure 1 foods-10-00940-f001:**
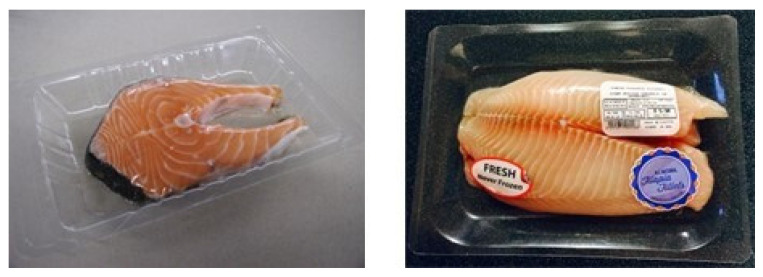
Application of vacuum skin packaging to seafood products (Packaging Europe) [74].

**Figure 2 foods-10-00940-f002:**
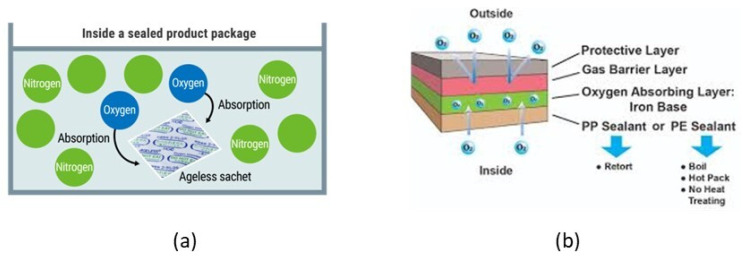
(**a**) Oxygen scavenger sachet, (**b**) oxygen scavenging film (ResearchGate) [101].

**Figure 3 foods-10-00940-f003:**
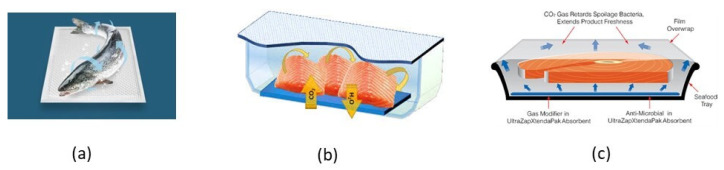
(**a**) CO_2_ emitter pad, (**b**) CO_2_ emitter and moisture absorber pad, (**c**) CO_2_ emitter and antimicrobial pad (ResearchGate) [101].

**Figure 4 foods-10-00940-f004:**
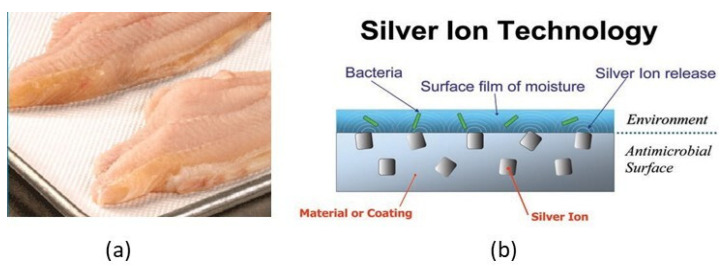
(**a**) Food-touch^®^ antimicrobial paper liner, (**b**) silver ion technology (Packaging Digest, Pharmaceutical Microbiology Resources) [103,104].

**Figure 5 foods-10-00940-f005:**
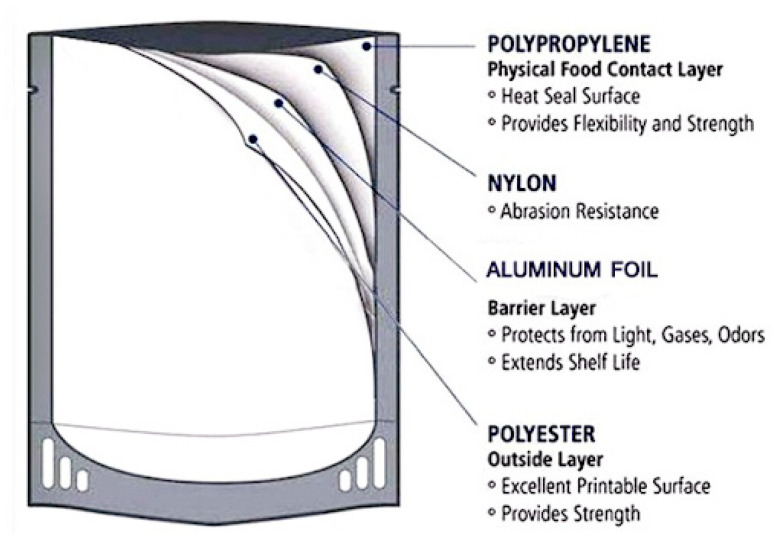
Retort multilayer pouch (Flair Packaging) [146].

**Figure 6 foods-10-00940-f006:**
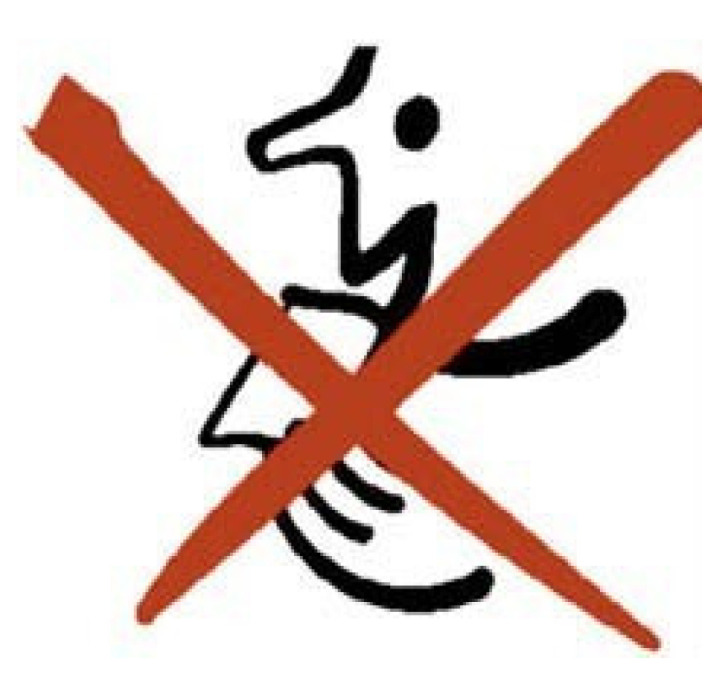
Symbol for ‘non-edible’ parts in active and intelligent material labeling (Regulation EC 450/2009) [96].

**Figure 7 foods-10-00940-f007:**
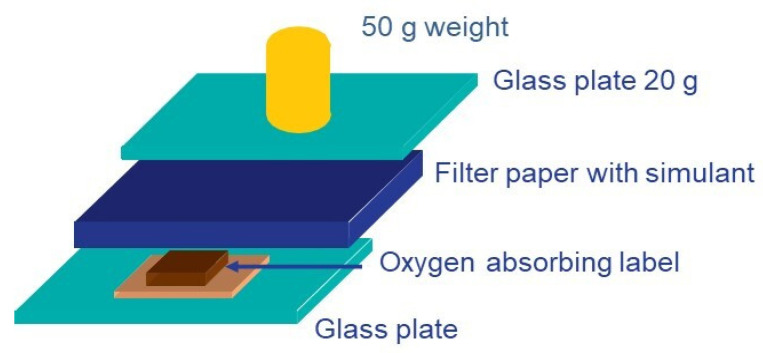
Graphical scheme of the dedicated migration test for oxygen scavenger (Dainelli et al.) [221]. Reprinted from [Elsevier], Copyright (2008), with permission from Elsevier.

**Table 1 foods-10-00940-t001:** Potential active packaging applications for seafood.

Τype of Active Packaging	Type of Food	Potential Benefit
Active scavenging systems		
Oxygen absorber	Fatty fish	Prevention of rancidity and discoloration
Moisture absorber	Fresh fish	Shelf life extension, reduction of moisture condensation within the package
Active releasing systems		
Antioxidant releaser	Fresh fatty fish	Enhancement of oxidative stability
Carbon dioxide emitter	Fresh fish	Shelf life extension, reduction in head space volume of MAP
Antimicrobial releaser	Fresh and smoked fish, fresh seafood	Retardation of microbial growth, shelf life extension

## Data Availability

Not applicable.

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
