# Peer review of "Recent Developments in Seafood Packaging Technologies"

_foods, 2021, doi:10.3390/foods10050940_

Round 1

Reviewer 1 Report

Line 124 (EO) should be defined because the abbreviation is used in the manuscript

Line 139 ¿total volatile count? Check if this is correct. In the manuscript the TVC is used for referring to Total Plate Counts of mesophilic microorganisms

Line 140 SSO or SSB, if use SSB take into account that must be changed in the whole text.

Line 199 Listeria monocytogenes only for the first time in the text, the followings must be written like L. monocytogenes. This is valid for all the names of microorganisms in the manuscript

Line 244 “The product is usually packaged in a barrier material along with a mixture of gases, the composition of which depends on the product properties, the expected shelf life and the storage conditions” Although you refer to a barrier material the composition of gases during storage also depends on barrier properties of the package. In line 249 the authors refer to this.

Line 273 “Clostridium botulinum” change by C. botulinum

Line 532 The authors should refer to species, because all the genus species (example Listeria) are not pathogens

568 “1000 mbar (1 atm) to 100 millibar” change by “1000 mbar (1 atm) to 100 mbar”

648 TBARS should be defined

749-919 It is a large description of “Active packaging”, It should be summarized because it diverts the content of the review into active packaging. References should only be made to those active packaging developments with applications to fish and fishery products because the Tittle of the Review is “Recent developments in seafood packaging technologies”

Line 990 Eschershia change by Escherichia

Line 1013 -1014 when reference to malondialdehyde is made, should be noted that it is the same that TBARS

Line 1203 and 1205 “Clostridium botulinum” change by C. botulinum

Line 1361 Reference of the maximum limits recommended should be provided

Line 1591 St. aureus or S. aureus, check the manuscript and always write it in the same form (line 1551, 1555)

Line 1610 CO2 or CO2, always write it in the same form

Line 1838 References

  • Scientific names in references must be in italic form
  • Reference 198, check if it is correct
  • There is only one 2020 reference, the most recent developments should be discussed. Example https://doi.org/10.3390/foods10020250

Author Response

Reviewer 1

comment

Line 124 (EO) should be defined because the abbreviation is used in the manuscript

Response  

The term (EO) is not mentioned anywhere close to l.124. It was found for the first time on l.359 were it was abbreviated.

comment

Line 139,  total volatile count? Check if this is correct. In the manuscript the TVC is used for referring to Total Plate Counts of mesophilic microorganisms

response

TVC stands for total viable count. See correction  l. 153

comment

Line 140 SSO or SSB, if use SSB take into account that must be changed in the whole text.

response

SSO stands for specific spoilage  organisms. See correction l.153-154

comment

Line 199 Listeria monocytogenes only for the first time in the text, the followings must be written like L. monocytogenes. This is valid for all the names of microorganisms in the manuscript

Response

Corrected.  See l. 241

comment

Line 244 “The product is usually packaged in a barrier material along with a mixture of gases, the composition of which depends on the product properties, the expected shelf life and the storage conditions” Although you refer to a barrier material the composition of gases during storage also depends on barrier properties of the package. In line 249 the authors refer to this.

Response

Corrected. See l.288

comment

Line 273 “Clostridium botulinum” change by C. botulinum

Response

Corrected. See l.315

comment

Line 532 The authors should refer to species, because all the genus species (example Listeria) are not pathogens

Response

Corrected. See l.571

comment

568 “1000 mbar (1 atm) to 100 millibar” change by “1000 mbar (1 atm) to 100 mbar”

Response

Corrected. See l.606

comment

648 TBARS should be defined

Response

Done. See l.686

comment

749-919 It is a large description of “Active packaging”, It should be summarized because it diverts the content of the review into active packaging. References should only be made to those active packaging developments with applications to fish and fishery products because the Tittle of the Review is “Recent developments in seafood packaging technologies”

Response

We have made an effort to shorten the ‘active packaging’ section. However in this section we basically present the principle of each active packaging technology. In the following sections we refer to the application of these techniques to seafood packaging.

comment

Line 990 Eschershia change by Escherichia

Response

Done see l.1040

comment

Line 1013 -1014 when reference to malondialdehyde is made, should be noted that it is the same that TBARS

Response

Done. See l.1064

comment

Line 1203 and 1205 “Clostridium botulinum” change by C. botulinum

Response

Done. See l.1266, 1268.

comment

Line 1361 Reference of the maximum limits recommended should be provided

Response

ICMSF, 1986. Microorganisms in Foods 2. Sampling for Microbiological Analysis: Principles and Specific Applications (2nd ed.), University of Toronto Press, New York.

Directive 95/149/EC, 1995. Fixing the total volatile basic nitrogen (TVB-N)

limit values for certain categories of fishery products and specifying the

analysis methods to be used. Official Journal of the European

Communities.

Above references have been included in the text. See l.1424.

comment

Line 1591 St. aureus or S. aureus, check the manuscript and always write it in the same form (line 1551, 1555)

Response

Done. See l. 1617, 1619, 1623, 1651, 1659.

comment

Line 1610 CO2 or CO2, always write it in the same form

Response

Corrected. See l.1678.

Line 1838 References

comment

Scientific names in references must be in italic form

Response

Corrected. See changes in red color in reference list

comment

Reference 198, check if it is correct

Response

Reference checked and corrected. See ref. 218

comment

There is only one 2020 reference, the most recent developments should be discussed. Example https://doi.org/10.3390/foods10020250

Response

We have included the suggested ref. as well as some other very recent references in the revised text. See l.181-200, 206-212, 771-801, 1129-1143, 1432-1437.

Reviewer 2 Report

The review is dealing with the important food issue. My comments are the following:

  1. The abstract is written too broadly; the aim should be clearly defined in the abstract same as the conclusion.
  2. The review is not having the introduction part. It should be written,
  3. Table 1 is containing well known information; it should be deleted. It seems that it is added only to fill the manuscript.
  4. Line 128: the part MICROBIAL SPOILAGE should be rewritten since it is not dealing at all with biogenic amines content. The following reference among others should be used: Đorđević, Đ., Buchtova, H., & Borkovcova, I. (2016). Estimation of amino acids profile and escolar fish consumption risks due to biogenic amines content fluctuations in vacuum skin packaging/VSP during cold storage. LWT-Food Science and Technology, 66, 657-663.
  5. Line 169: the part OXIDATION AND HYDROLYSIS should be updated with the information about primary and secondary products of oxidation. Certain table can be made.
  6. Vacuum skin packaging should be a separate chapter.
  7. Line 734: this chapter is not necessary to be alone. It should be merged with the previous chapter.
  8. Figure 3: is there reference for it?
  9. Figure 4: is there reference for it?
  10. Figure 5: is there reference for it?
  11. Freshness indicators: more research about pH indicators edible films should be included.
  12. Figure 6: there is no reference.
  13. Line 1228: missing reference.
  14. The manuscript is missing the conclusion part.
  15. Authors should also consider to perform certain meta analysis.

Author Response

Reviewer 2

Top of Form

comment

  1. The abstract is written too broadly; the aim should be clearly defined in the abstract same as the conclusion.

Response

the Abstract has been revised  clearly defining the aim of the work. A conclusion was also added to the text. See l.10-23, 1873 and 1885-1888.

comment

  1. The review is not having the introduction part. It should be written,

Response

A short Introduction has been added to the text. See l.26-41.

comment

  1. Table 1 is containing well known information; it should be deleted. It seems that it is added only to fill the manuscript.

Response

Table 1 has been deleted as suggested by the reviewer.

comment

  1. Line 128: the part MICROBIAL SPOILAGE should be rewritten since it is not dealing at all with biogenic amines content. The following reference among others should be used: Đorđević, Đ., Buchtova, H., & Borkovcova, I. (2016). Estimation of amino acids profile and escolar fish consumption risks due to biogenic amines content fluctuations in vacuum skin packaging/VSP during cold storage. LWT-Food Science and Technology, 66, 657-663.

Response

A short paragraph has been added to the text on Biogenic amines. See l. 181-197. The article by Dordevic et al., (2016) has been used in the chapter on Vacuum Packaging. See l. 787-793.

comment

  1. Line 169: the part OXIDATION AND HYDROLYSIS should be updated with the information about primary and secondary products of oxidation. Certain table can be made.

Response

We have updated information on secondary oxidation products as suggested by the reviewer. See l. 206-212, and 215-217.

comment

  1. Vacuum skin packaging should be a separate chapter.

Response

 Since Vacuum skin packaging is a type of vacuum packaging we believe there is no need to have a separate chapter for the former.

comment

  1. Line 734: this chapter is not necessary to be alone. It should be merged with the previous chapter.

Response  

The only reason why we have placed this small chapter separately is that we distinguished fish from fishery products preservation exactly as done for MAP. In essence there is no difference between the two. Just for homogeneity purposes we have retained this chapter separately.

comment

  1. Figure 3: is there reference for it?

Response

Reference has been added. See l. 894.

comment

  1. Figure 4: is there reference for it?

Response

Fig 4 has been deleted  based on the suggestion of reviewer 1.

comment

  1. Figure 5: is there reference for it?

Response

Figure 5 is now Figure 4 in the revised text. References have been added. See l.942.

comment

  1. Freshness indicators: more research about pH indicators edible films should be included.

Response

Additional information on pH sensitive, freshness indicators has been included in the texty. See l.1129-1143.

comment

  1. Figure 6: there is no reference.

Response

Figure 6 is now Figure 5 in the revised text. Reference has been added. See l. 1293.

comment

  1. Line 1228: missing reference.

Response  text between l.1254 and 1262 (of the revised text) refers to ref. Tsironi et al.[141] already provided.

comment

  1. The manuscript is missing the conclusion part.

Response

We have slightly modified the “Future outlook ‘ section to also serve as a ‘Conclusion ‘ section. See l.1874-1900.

comment

  1. Authors should also consider to perform certain meta analysis.

Response

‘A meta-analysis is a statistical analysis that combines the results of multiple scientific studies. Meta-analysis can be performed when there are multiple scientific studies addressing the same question, with each individual study reporting measurements that are expected to have some degree of error’.

Based on this ‘definition’ of meta-analysis  we do not believe it is feasible to run such a statistical analysis on data presented in the present work since we do not have access to respective statistical data.

Round 2

Reviewer 2 Report

I have no further comments.